# Spatio–temporal averaging of jets obscures the reinforcement of baroclinicity by latent heating

Henrik Auestad[1], Clemens Spensberger[2], Andrea Marcheggiani[2], Paulo Ceppi[3], Thomas Spengler[2], and Tim Woollings[1]

[1]Atmospheric, Oceanic and Planetary Physics, University of Oxford, Oxford, UK
[2]Geophysical Institute, University of Bergen, and Bjerknes Centre for Climate Research, Bergen, Norway
[3]Department of Physics, Imperial College London, London, UK

**Correspondence:** Henrik Auestad (henrik.auestad@physics.ox.ac.uk)

**Abstract.** Latent heating modifies the jet stream by modifying the vertical geostrophic wind shear, thereby altering the potential for baroclinic development. Hence, correctly representing diabatic effects is important for modelling the mid-latitude atmospheric circulation and variability. Yet, the direct effects of diabatic heating remain poorly understood. For example, there is no consensus on the effect of latent heating on the cross-jet temperature contrast. We show that this disagreement is attributable to the choice of spatio–temporal averaging. Jet representations relying on averaged wind tend to have the strongest latent heating on the cold flank of the jet, thus weakening the cross-jet temperature contrast. In contrast, jet representations reflecting the two-dimensional instantaneous wind field have the strongest latent heating on the warm flank of the jet. Furthermore, we show that latent heating primarily occurs on the warm flank of poleward directed instantaneous jets, which is the case for all storm tracks and seasons.

## 1 Introduction

Jet streams are an integral part of the planetary scale flow, guiding synoptic weather systems and leading to the existence of storm tracks (Chang et al., 2002; Martius et al., 2010; Wirth et al., 2018). In return, these synoptic weather systems shape the planetary scale flow by redistributing heat, momentum, and vorticity (Edmon et al., 1980; Hoskins et al., 1983) and self-maintain the baroclinicity in the storm track through diabatic heating (Hoskins and Valdes, 1990; Papritz and Spengler, 2015). While diabatic heating related to moist processes climatologically maintains baroclinicity (Papritz and Spengler, 2015; Marcheggiani and Spengler, 2023), there is conflicting evidence to which extent it dampens or amplifies low frequency variability (Xia and Chang, 2014; Woollings et al., 2016; Lutsko and Hell, 2021). We disentangle these conflicting findings by contrasting jet-relative diabatic heating for different jet representations.

On synoptic time scales, latent heating amplifies cyclone growth (see Wernli and Gray, 2023, for a review) as well as enstrophy (Black, 1998) and influences the jet stream through changes in potential vorticity gradients (Harvey et al., 2020; Bukenberger et al., 2023). Furthermore, as latent heating modifies baroclinicity, it also affects baroclinic wave activity on time scales beyond the lifetime of a single cyclone (Hoskins and Valdes, 1990; Papritz and Spengler, 2015; Weijenborg and Spengler, 2020). As hypothesised by Robinson (2000), anomalous cyclone activity and eddy momentum fluxes following from

anomalous baroclinicity may modify the persistence of the planetary scale flow (e.g. Lorenz and Hartmann, 2001; Blanco-Fuentes and Zurita-Gotor, 2011; Zurita-Gotor et al., 2014; Lorenz, 2022). Thus, if latent heating dampens the anomalous baroclinicity, it should also dampen the planetary scale flow variability, and vice-versa. While this mechanism concerns the amplitude of momentum fluxes, the resulting baroclinicity may also affect the preferred type of wave breaking and therefore the direction of the momentum fluxes (Orlanski, 2003; Rivière, 2009).

Xia and Chang (2014) conditioned zonal mean latent heating (precipitation) on the first principal component (PC1) of the 300 hPa daily zonally averaged zonal wind in the Southern Hemisphere and found it to weaken the cross-jet temperature contrast in both phases of PC1. Following a similar argument, Bembenek et al. (2020) and Lutsko and Hell (2021) showed that latent heating mainly occurs on the poleward flank of a zonal mean jet, abating the mean baroclinicity and therefore weakening the mean westerlies. The experiment of Yamada and Pauluis (2017) adds further nuance, showing that latent heating occurs in the core of a zonal mean jet for a cyclonic wave breaking event. On the other hand, Woollings et al. (2016) analysed the instantaneous two-dimensional tendency of the isentropic slope (Papritz and Spengler, 2015), showing a positive feedback to baroclinicity in the anomalous position of the storm track. The isentropic slope framework also shows that latent heating maintains the seasonal mean baroclinicity in the North Atlantic storm track (Papritz and Spengler, 2015; Marcheggiani and Spengler, 2023).

The disagreement between Xia and Chang (2014) and Woollings et al. (2016) could stem from the choice of storm track region and season. Xia and Chang (2014) studied the Southern Hemisphere during summer, while Woollings et al. (2016) studied the North Atlantic during winter. However, the studies also differ in their methods. Xia and Chang (2014) used zonally averaged daily mean wind and precipitation and assumed that latent heating is linearly dependent on PC1 of the zonal mean wind. Thus, their results could be sensitive to their choice of jet definition and spatio–temporal averaging. Woollings et al. (2016), on the other hand, used the isentropic slope framework which requires neither temporal nor spatial averaging and does thereby not rely on zonal means.

To address these conflicting results, we compile jet conditional climatologies of latent heating and other pertinent variables to explore their sensitivity to spatio–temporal averaging and different jet definitions, either based on zonal means or the two-dimensional wind field. We specifically explore the extent to which latent heating occurs on the warm or cold flank of each jet definition, reflecting on the fact that *jet* is a widely used term in the literature. Climatologists often refer to 'mean flow' features as jets, while synopticians tend to reserve the term for instantaneous jet–like wind profiles.

## 2 Data

We use three-hourly data from ERA5 (Hersbach et al., 2020) for December, January, and February (DJF) for the period 1979 to 2020. We focus on the North Atlantic but also show some analysis for the North Pacific, South Pacific and Indian Ocean. We use the three-dimensional wind, specific humidity and temperature on a $0.5°$ horizontal grid and 22 pressure levels [10, 30, 50, 100, 150, 200, 250, 300, 350, 400, 450, 500, 550, 600, 650, 700, 750, 800, 850, 900, 950, 1000] hPa. Diabatic temperature tendencies due to all physical parameterizations as well as only due to radiation are available on model levels and interpolated

**Table 1.** The five different jet representations.

| Label | Full name |
|-------|-----------|
| 3h-Z | Three-hourly sector-mean jet |
| 1d-Z | Daily sector-mean jet |
| 10d-Z | 10-day low-pass sector-mean jet |
| 3h-2D | Three-hourly two-dimensional jet |
| 10d-2D | 10-day low-pass two-dimensional jet |

to the same pressure levels. We also use horizontal wind on the 2 PVU ($2 \times 10^{-6}$ K kg$^{-1}$ m$^2$ s$^{-1}$) surface as well as large-scale and convective precipitation.

We subtract the radiative diabatic tendencies from the total diabatic temperature tendency due to physical parametrization and assume that the remaining diabatic heating tendencies are dominated by latent heating above the boundary layer (Papritz and Spengler, 2015). Following the example of Papritz and Spengler (2015), we average the "diabatic minus radiative" heating over 700–200 hPa and refer to this as latent heating. Correspondingly, the vertical wind and the specific humidity are also averaged over 700–200 hPa.

## 3 Jet detection

We consider two types of jet representations (Table 1): one based on zonally averaging the wind field, referred to as *sector-mean* jets, and the other based on jet axes defined by the two-dimensional wind field, which we refer to as *two-dimensional* jets. Zonally averaged wind is frequently used to identify and diagnose the mid-latitude jet stream (e.g. Robinson, 2000; Lorenz and Hartmann, 2001; Woollings et al., 2010; Blanco-Fuentes and Zurita-Gotor, 2011; Zurita-Gotor et al., 2014; Bembenek et al., 2020; Lutsko and Hell, 2021). This has the advantages of efficiency and a clear connection to the zonal momentum budget, but with the disadvantage that zonally averaging neglects the meandering characteristic of the mid-latitude jet (Spensberger et al., 2023). The two-dimensional jet representation, on the other hand, captures such waviness.

The sector-mean jet position is computed by picking the latitude of the strongest zonally averaged zonal wind between $60°$W and $0°$E on the 2 PVU surface. We perform this analysis for every time step to represent the jet over the central to eastern North Atlantic Ocean, only allowing for one jet latitude for each time step. The method is similar to Woollings et al. (2010), but using upper- instead of lower tropospheric zonal wind.

The two-dimensional jet representation follows Spensberger et al. (2017). In short, jets are detected where the wind shear perpendicular to the wind direction is zero on the 2 PVU surface.

$$\frac{\partial U}{\partial n} = 0, \tag{1}$$

where $U = \sqrt{u^2 + v^2}$ and $n$ is perpendicular to the wind direction. Like Spensberger et al. (2017), we apply a T84 truncation to the data before detecting the jets to smooth out smaller scale features and provides for more spatially coherent jets. We use the freely available Python library Dynlib (Spensberger, 2024) for the computations.

The two-dimensional jets detected within [80° W–10° E, 20° N–70° N] are used to represent the North Atlantic. An undesirable effect of Eq. 1 is that it is also satisfied for wind speed minima. Additionally, high wind speed areas do not always have a well defined jet structure. We mitigate these challenges by only considering locations where $K \leq (\partial U/\partial n)^2 + U \partial^2 U/\partial n^2$ (Spensberger et al., 2017). This approach only identifies wind speed maxima as jets. It also allows for weaker winds with a clear jet like profile to be classified as jets and filters out points of high wind speed without a well defined jet structure. $K$ is set to the 12.5th percentile of $(\partial U/\partial n)^2 + U \partial^2 U/\partial n^2$. For the three-hourly data, $K = -0.732 \cdot 10^{-8} \mathrm{s}^{-2}$, identical to Spensberger et al. (2023). Note that generally several two-dimensional jets are detected at each time step. See Spensberger et al. (2017) for more details on the jet axis detection.

To assess the sensitivity of our findings to temporal averaging, we compute jet latitudes using instantaneous three-hourly, daily mean, and 10-day low-pass filtered data. For the 10-day low-pass filtered data, daily mean values are first computed from the three-hourly data before applying a Lanczos filter (Duchon, 1979) with a 61-day window. The two-dimensional jets are computed on the three-hourly and the 10-day low-pass filtered wind field. For the two-dimensional jet detection using the 10-day low pass data, we apply $K = -1.044 \cdot 10^{-10} \mathrm{s}^{-2}$.

We refer to the sector mean jets as 3h-Z, 1d-Z, and 10d-Z for the three-hourly, daily, and 10-day low-pass wind, respectively (Table 1). The two-dimensional jets are analogously referred to as 3h-2D and 10d-2D. Illustrating the differences between the jet definitions, Fig. 1a shows the sector mean jets simply intersecting the latent heating associated with the two cyclones on 14 January 2009 at 2100 UTC. The 3h-Z jet is located further south compared to both 1d-Z and 10d-Z. In contrast, the 3h-2D jets exhibit synoptic scale variations, outlining troughs and ridges (Fig. 1b). The 10d-2D version does not pick up any synoptic variability but rather follows a straight line from the Gulf Stream region to Northern Europe.

## 4 Construction of the jet conditional climatologies

We compile climatologies for vertical wind, specific humidity, latent heating, and precipitation for the different jet definitions. All variables are of three-hourly time resolution, irrespective of the time filtering used for the jet definition.

For climatologies conditional on the sector-mean jet latitude, we first average the variables over the same longitudinal sector (60° W–0° E) as used to define the sector-mean jet latitude (Fig. 2a). The resulting sector mean values are then capped at 20° latitude ($\sim 2200 \, \mathrm{km}$) on each side of the concurrent three-hourly, daily, and 10-day low-pass sector-mean jet. The data are organised by the latitude of the concurrent sector-mean jet, given on the horizontal axis in the right panel of Fig. 2a, so that the lowest jet latitude states are on the left and the highest jet latitude states on the right. The vertical axis in the right panel of Fig. 2a shows the cross-jet distance spanning $\pm 20°$ latitude. Finally, the data around jets found at similar latitudes are then averaged over by binning the data using 30 bins in the horizontal and 20 bins in the vertical direction, before averaging over each bin.

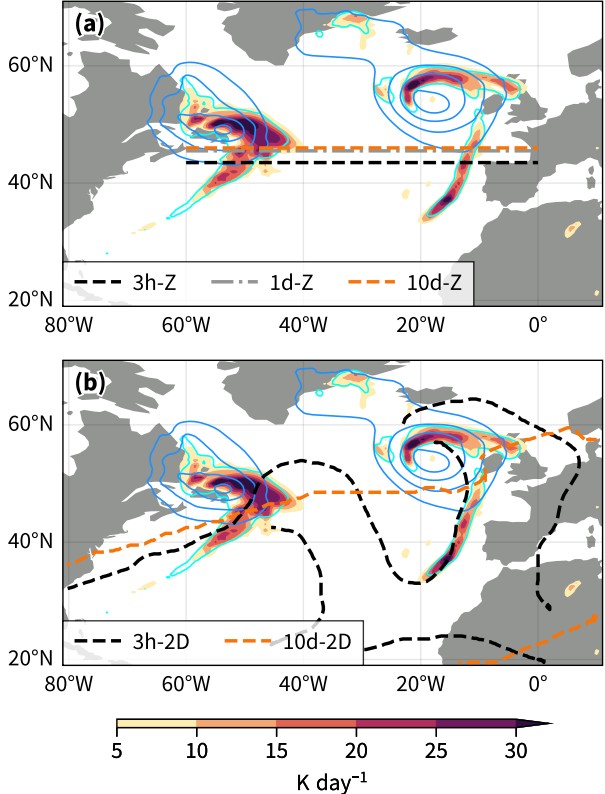

**Figure 1.** Latent heating (shading), large-scale precipitation (cyan; $0.33 \, \mathrm{mm \, h^{-1}}$) and sea level pressure (blue; 960, 970, 980, 990, 1000 hPa) for 14 January 2009 at 2100 UTC. The dashed lines in (a) show the sector-mean jets computed on three-hourly zonal wind (black; 3h-Z), daily mean zonal wind (grey; 1d-Z), and 10-day low-pass zonal wind (orange; 10d-Z). The dashed lines in (b) show the two-dimensional jets computed on three-hourly winds (black; 3h-2D), and 10-day low-pass winds (orange; 10d-2D).

The computation of climatologies conditional on the latitude of the two-dimensional jets is illustrated in Fig. 2b. A random set of 100,000 cross sections are sampled from all two-dimensional jets (also used by Spensberger et al., 2023), as exemplified by the black arrows. The variables are interpolated to the cross sections, spanning 500 km on each flank of the jets normal to the jet direction. The cross sections are then organised by the latitude of the jet core in each respective cross section. Given that the two-dimensional jets can take any direction in the latitude–longitude space, we rotate the two-dimensional jets so that they appear as pure westerlies. With this rotation, the left flanks of the jets are on the positive- and the right flanks on the negative side of the vertical axis. By the thermal wind relation, the left flank corresponds to the cold flank of the jet, while the right flank correspond to the warm flank of the jet. The data are coarse grained by binning the data using 20 bins in the horizontal and 10 bins in the vertical direction before averaging over each bin.

Lastly, we compute climatologies conditional on the direction of the two-dimensional jets (see Fig. 2c). As for the other conditional climatologies, we make use of jet cross sections but this time organise them by the direction of the jet rather than

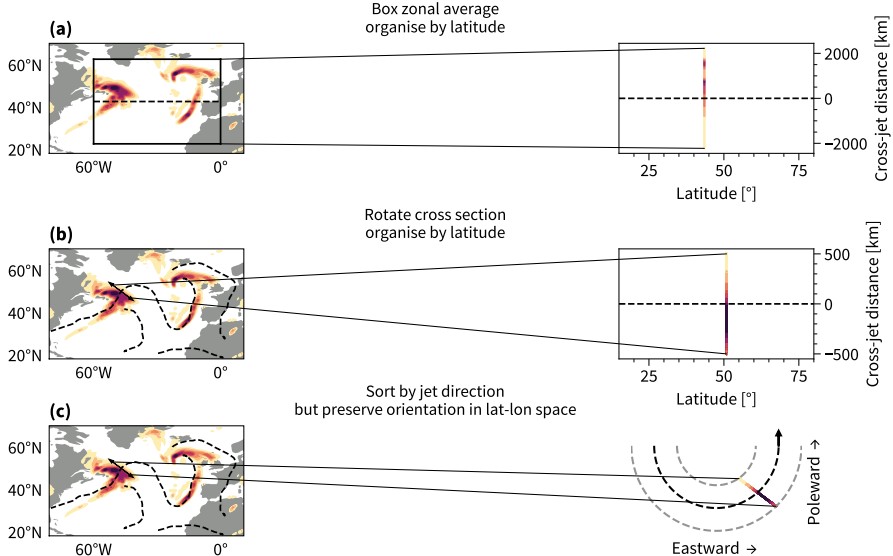

**Figure 2.** Illustration of how we compile the climatologies conditional on jet latitude based on (a) sector-mean jet latitudes, (b) latitude of the two-dimensional jets, and (c) jet direction. The dashed lines illustrate the jets. The box illustrates the area for the sector zonal averaging and the arrows illustrate the jet cross section.

their latitude. Their original orientation in the latitude–longitude space is preserved and the cross sections are thus organised in semicircles according to their orientation. The climatologies conditional on jet direction have been coarse grained by binning and averaging the data, here using 30 bins in both the horizontal and vertical direction. The number of bins scales with how the data are distributed in the three climatologies.

The binning and averaging gives a smoothing effect and the climatologies thus show the estimated expected value of each variable around the jets, conditional on the latitude and direction of the jets. The spread in the data is discussed at the end of Sect. 6. The results are qualitatively the same when increasing the number of bins.

We focus on eddy-driven jets, but the lowest latitude jets appearing in our analyses are likely of subtropical nature (Lee and Kim, 2003). This is the case for both the low-pass and the instantaneous jets (Spensberger et al., 2023). We do not exclude subtropical jets, as we do not have a consistent criterion to do so for both the sector-mean jet detection and the two-dimensional jet detection.

## 5 Latent heating across sector-mean jets

The strongest sector mean winds are found around the respective jet latitudes (Fig. 3 a–c) with the pattern in the wind being somewhat smoother and more confined around the 3h-Z compared to the time filtered 1d-Z and 10d-Z jet latitudes. The sector mean winds do, however, show quite large values in a broad latitudinal band around the detected sector-mean jet. This is not

unexpected as we compute sector averages over wavy and often tilted jets. The sector mean wind field might also feature several local maxima, which would contribute to the breadth of the wind profile around the identified jet latitude.

As expected from the thermal wind relation, the poleward flanks of the sector-mean jets are colder than the equatorward flanks (Fig. 3 a–c). Additionally, few 10d-Z jets are detected poleward of 65°N (see Appendix A), causing the volatility in the temperature contours there. Due to the small sample size of 10d-Z jets poleward of 65°N, the patterns there might not be robust and should be interpreted with care.

Two distinct patterns in vertical wind are visible around the sector-mean jets, with one of them being sensitive to the time 145 filtering. The main pattern of vertical wind, which is independent of the time filtering, resembles the Ferrel cell, with descent observed around 30° N and ascent further poleward (Fig. 3d–f). Around the 3h-Z jets poleward of 40° N, we also notice a secondary pattern in the vertical wind featuring ascent on the warm flank of the jet. The same pattern is visible, but somewhat diluted, for the 1d-Z and only barely visible for the most poleward 10d-Z jets. Hence, the secondary pattern is sensitive to the time filtering.

The distribution of specific humidity, in comparison to vertical wind, is less affected by the time filtering with higher values on the warm flank compared to the cold flank for all the sector-mean jets (Fig. 3g–i).

As mentioned, sector averaging imposes a smoothing effect similar to time filtering, as it averages over several Rossby-wave phases and cyclones. Given the longitudinal confinement of the sector, the zonal distance of averaging decreases with latitude. Hence, the effect of filtering is larger at lower latitudes, which may explain why the zonal wind profile and cross-jet contrast 155 in the vertical wind sharpens at higher latitudes (Fig. 3a–f).

Condensation and thus latent heating typically occur in regions of ascent where moisture is available. Irrespective of time filtering, the sector-mean jets equatorward of $\sim 45°$ N exhibit more latent heating on the poleward flank compared to the equatorward flank (Fig. 3j–k). For jets poleward of $\sim 40°$ N, latent heating follows the aforementioned secondary pattern in vertical wind, which is sensitive to time filtering. As a result, a clear cross-jet contrast in latent heating is seen without time 160 filtering, such that the heating maximum occurs on the equatorward flank of jets poleward of $\sim 40°$ N. At the same latitudes, there is no clear cross-jet contrast in neither vertical wind nor latent heating around the 10d-Z jets.

The sensitivity of the latent heating pattern is congruent with the large-scale precipitation (Fig. 4), which spreads out over all latitudes characterised by ascent (Fig. 3), while also forming a trail of stronger precipitation on the equatorward flank of the jets poleward of $\sim 40°$ N. Convective precipitation, on the other hand, is first and foremost located on the poleward flank 165 of the sector-mean jets, where the time filtering only slightly affects the pattern (Fig. 4). The convective precipitation features a banded latitudinal structure which indicates that it has a strong geographical preference. The larger values around 35–40° N match the climatologically large values of convective precipitation over the Gulf Stream and the cutoff at 60° N coincides with little convective precipitation north of Iceland within the domain of sector zonal averaging (60° W–0° E; not shown).

The relationship we find between the sector-mean jets and latent heating is similar to Xia and Chang (2014). They regressed 170 observed precipitation onto the zonal index in the Southern Hemisphere and found it to follow the poleward flank of the zonal mean jet anomalies. They hence inferred that latent heating weakens the baroclinicity in the anomalous jet. Xia and Chang (2014) use daily data and their finding is consistent with the sector-mean jets equatorward of 40° N (Fig. 3k), given that

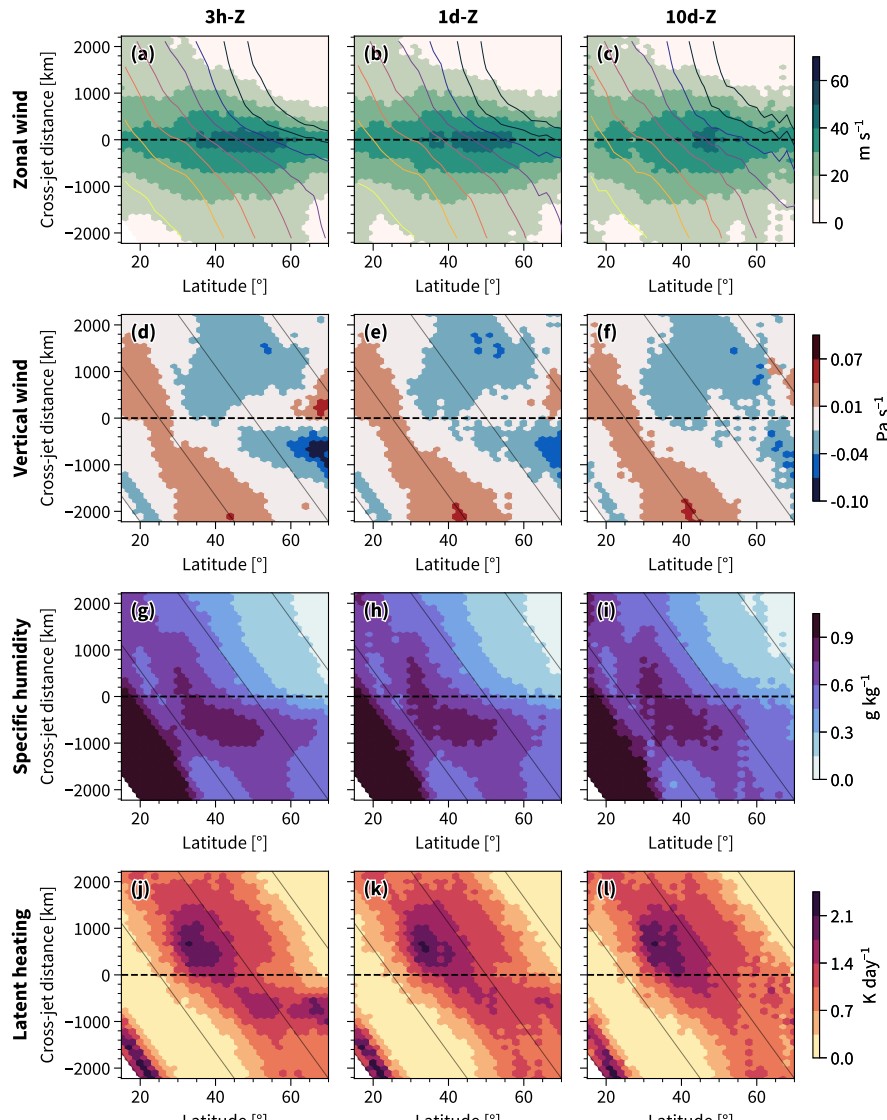

**Figure 3.** Three-hourly zonal wind (2 PVU; a–c), vertical wind (700–200 hPa; d–f), specific humidity (700–200 hPa; g–i) and latent heating (700–200 hPa; j-l) averaged from 60° W to 0° E and shown against the latitude of the concurrent 3h-Z jet (left), 1d-Z jet (middle), and 10d-Z jet (right). The jet core is shown by the black dashed line. The grey diagonal lines show constant latitudes for 0, 25, 50, 75 ° N. The contours in (a-c) show potential temperature at 500hPa, with intervals 290 (yellow), 295, 300, 305, 310, 315, 320, 325 (black) K.

our results hold for the Southern Hemisphere during summer. Similarly, Bembenek et al. (2020) and Lutsko and Hell (2021) revealed the same negative effect on baroclinicity as deduced from the jets between 30° N–45° N (Fig. 3) when analysing the location of precipitation relative to a zonal mean jet. The 10d-Z analysis also resemble Lachmy and Kaspi (2020), who used

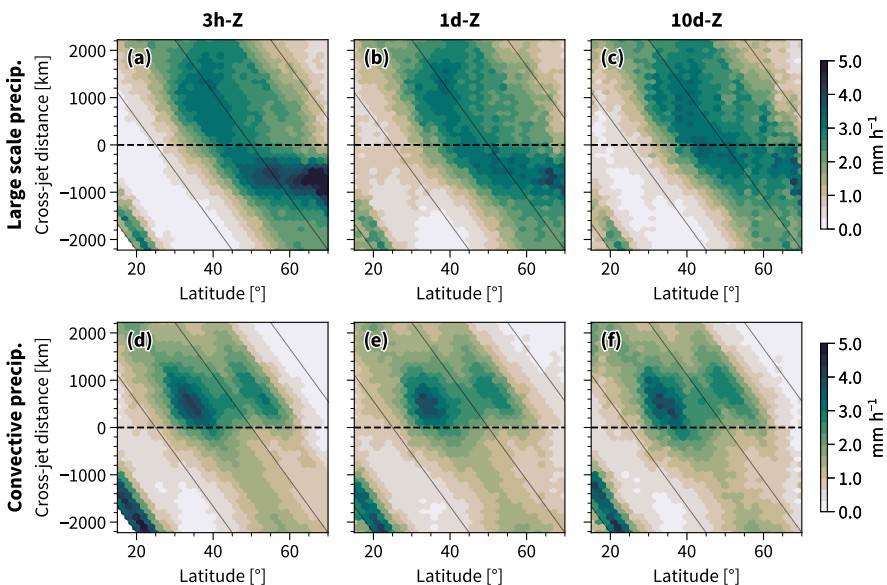

**Figure 4.** Like Fig. 3 but for large-scale precipitation (a–c) and precipitation from the ERA5 convection scheme (d–f).

monthly and zonally averaged values to show that the *eddy driven jet* is shifted equatorward of the largest eddy heat fluxes in winters of strong latent heating. During winters of weak latent heating, they show that the eddy driven jet and the largest eddy heat fluxes are closer in latitude.

## 6   Latent heating around two-dimensional jets

As expected, the wind in the direction of the jet concentrates around the 3h-2D jets, with the wind profile being somewhat sharper on the poleward flank compared to the equatorward flank, exhibiting a clear cross-jet contrast in temperature (Fig. 5a). The wind is far less concentrated around the 10d-2D jets and has relatively low values for the most poleward located jets, exhibiting a weak cross-jet temperature contrast (Fig. 5b). In other words, the position of the 10d-2D jet is a poor diagnostic for the position of the instantaneous wind.

The distribution of vertical wind around the two-dimensional jets is clearly affected by time filtering. Updrafts are located on the warm flank and downdrafts on the cold flank of the 3h-2D jet axes (Fig. 5c). This is in striking contrast to the sector-mean approach (Fig. 3), where the Ferrel cell structure comprises the opposite pattern. It also contrasts the vertical wind distribution around the 10d-2D jets, which appears insensitive to the presence of the jets (Fig. 5d) and similar to the time filtered sector-mean jets, ascent appears to be mainly a function of the jet latitude.

Time filtering has a similar effect on specific humidity. Specific humidity is higher on the warm side for the 3h-2D jets (Fig. 5e), while the specific humidity distribution is insensitive to the presence of 10d-2D jets and, again, is mostly a function of the jet latitude (Fig. 5f).

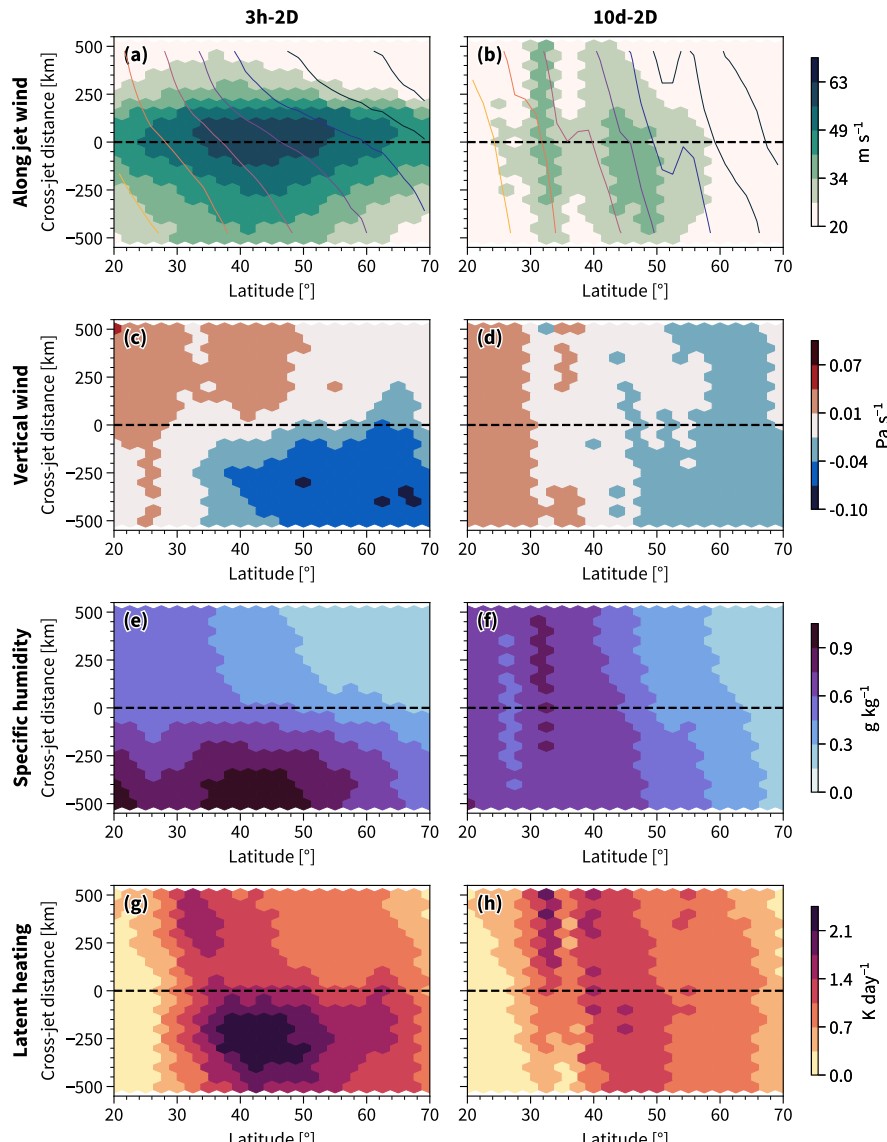

**Figure 5.** Three-hourly winds in the direction of the jet (2 PVU; a–b); vertical wind (700–200 hPa; c–d), specific humidity (700–200 hPa; e–f) and latent heating (700–200 hPa; g–h) interpolated to the cross sections sampled from the two-dimensional jets: 3h-2D (left) and 10d-2D (right). Data are plotted against the latitude of the jet core of each cross section. The warm flanks of the jets lie below- and the cold flanks above the black dashed line, which denotes the jet core.

Latent heating is largest in the region of ascent and higher values of specific humidity (Fig. 5g). In general, more latent heating occurs on the warm flank compared to the cold flank of the 3h-2D jets poleward of 30° N. Some more substantial latent heating on the cold side is seen for jets around 35° N, although still weak in comparison to the warm flank. Similar to vertical

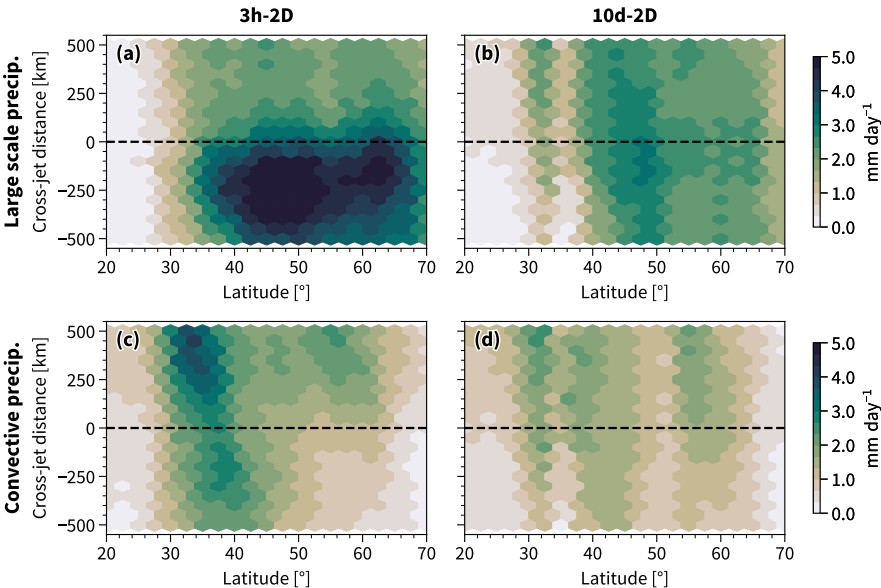

**Figure 6.** Like Fig. 5 but for large-scale precipitation (a–b) and precipitation from the ERA5 convection scheme (c–d) conditional on the 3h-2D jet axes (left) and the 10d-2D jet axes (right).

motion and specific humidity, latent heating appears independent of the presence of the 10d-2D jets with no clear cross-jet contrast in latent heating visible (Fig. 5h). The insensitivity in vertical wind, specific humidity and latent heating likely follows from the 10d-2D jets poorly representing instantaneous winds and therefore also poorly representing a barrier between cold and warm air. Thus, the 10d-2D jets will be out of phase with the dynamics that drives latent heating.

Similar to the sector-mean jets, it is primarily the large-scale precipitation that changes with time filtering around the two-dimensional jets (Fig. 6). Large-scale precipitation concentrates on the warm flank of the 3h-2D jets (Fig. 6a), though, similar to latent heating, has no preference for either flank of the 10d-2D jets (Fig. 6b). The large-scale precipitation is generally stronger than the convective precipitation in the two-dimensional jet conditional climatologies. Interestingly, the convective precipitation is co-located with the somewhat more substantial latent heating on the cold flank of the 3h-2D jets around 35° N.

Although more convective precipitation is expected around the 3h-2D jets, the convective precipitation pattern is qualitatively similar in 3h-2D and 10d-2D (Fig. 6c,d). This suggests that the time-filtering has less influence on the qualitative relationship between the jet and convective precipitation, compared to large-scale precipitation. For both the 3h-2D and the 10d-2D jets, the convective precipitation is stronger on the cold flank compared to the warm flank, although the cross-jet contrast is subtle for the 10d-2D. The latitudinal stationarity of this signal implies that it is linked to a geographical feature, like the Gulf Stream.

Overall, the distribution of latent heating conditional on the two-dimensional jets is strikingly different to that found by Xia and Chang (2014), Bembenek et al. (2020), and Lutsko and Hell (2021). This has profound repercussions for the interpretation of the effect of latent heating on baroclinicity and jet positioning. Whereas previous studies highlighted detrimental effects of

latent heating on the jet, in both the climatological and anomalous positions of the jet, our results based on instantaneous and two-dimensional jets indicate an enhancement of baroclinicity at all latitudes in the extratropics.

## 7   Latent heating conditional on jet direction

The relationship between jet and latent heating is sensitive to the instantaneous jet direction (Fig. 7), suggesting that the two are linked on synoptic time- and spatial scales. As expected from the Clausius–Clapeyron relation, specific humidity is generally higher on the warm flank compared to the cold flank (Fig. 7b). In contrast, ascent and latent heating are strongest along the warm flank of poleward oriented 3h-2D jets (Fig. 7a,c). Equatorward oriented jets are associated with descending motion and very low heating rates. Altogether, this is consistent with the synoptic picture of latent heating occurring in the warm, moist air flowing poleward and upward in warm conveyor belts, producing negative PV on the equatorward flank of jet (Madonna et al., 2014; Harvey et al., 2020). Figure 7 only shows westerly jets, which make up 90 percent of the 3h-2D jets detected in the NA. Easterly 3h-2D jets are discussed in Appendix B.

The location of latent heating relative to the jet core yields a strengthening of the baroclinicity for poleward oriented jets. The increase of baroclinicity (slope of isentropic surfaces) normal to the jet as a consequence of latent heating is confirmed when analysing the diabatic tendency of the isentropic slope (not shown).

The sensitivity of a given variable to the 3h-2D jet direction most likely explains the sensitivity to the spatio–temporal averaging shown above. Given that specific humidity and convective precipitation show similar distributions for all jet directions (Fig. 7b,f), they appear less sensitive to the averaging. Vertical motion (Fig. 7a), latent heating (Fig. 7c), and large-scale precipitation (Fig. 7e), in contrast, are highly sensitive to the orientation of the jet. Thus, when averaging in time and space, the heating appears at a relatively higher latitude compared to the average jet position, even though the heating occurs on the warm flank of the instantaneous jet.

The Lorenz framework is often used to understand baroclinic energetics, by separating the flow into mean and eddy components. While diabatic effects weaken the midlatitude mean available potential energy (APE) relative to a global reference state (Novak and Tailleux, 2018), latent heating is expected to increase APE relative to a local reference state (Danard, 1966). For latent heating, we showed that a split into mean and eddy components might be misleading, likely due to the asymmetric nature of latent heating with regard to the flow. This is in line with Danard (1966) who argues that the heating occurs in anomalously warm air and strengthens the large scale horizontal temperature gradients in the lower and middle troposphere. Also the isentropic slope budget of Papritz and Spengler (2015) shows that latent heating contributes positively to the overall slope/baroclinicity, and the resulting slope may outlive the original cyclone (Weijenborg and Spengler, 2020).

The relationship between latent heating and the 3h-2D jets holds for all the major storm track regions in their respective winters (Figs. 8, 9) as well as for the other seasons (not shown). Figures 8, 9 show the mean latent heating across the 3h-2D jets. It is important to note that although the strongest latent heating most often lies on the warm flank of the two-dimensional instantaneous jets, there are still many cases where the reverse is true, which is discussed in Appendix A.

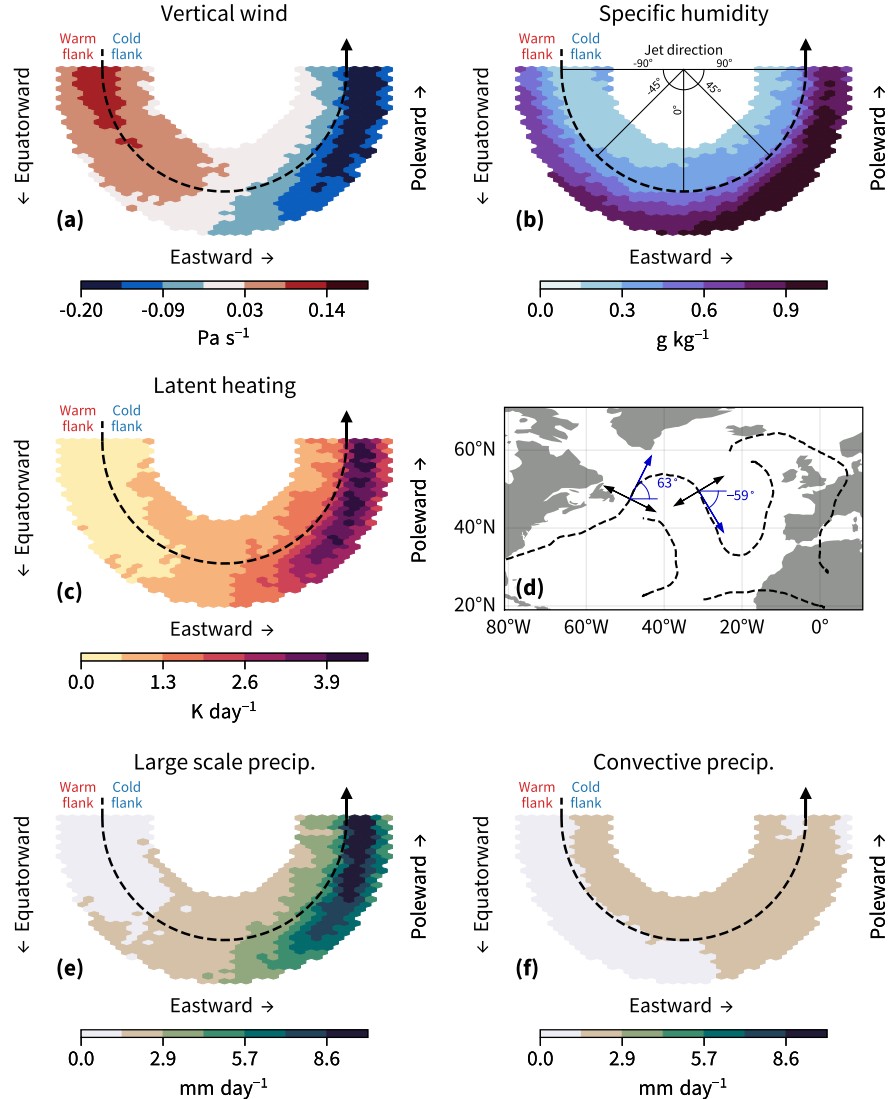

**Figure 7.** Cross sections for vertical wind (a; 700–200hPa); specific humidity (b; 700–200hPa); latent heating (c; 700–200hPa); large-scale precipitation (e); and precipitation from the convection scheme in ERA5 (f) for the 3h-2D jets. The cross sections are sorted by the jet direction and shown with their original orientation in the longitude–latitude plane. The jet directions are parallel to the orientation of the jet core (black dashed line). The angles given in (b) refer to the jet direction. Jet directions with $> 0°$ have a poleward component, with $= 0°$ are perfectly zonal, and with $< 0°$ have an equatorward component. Jet directions of two illustrative cross sections are shown in blue in panel (d). Only jet directions between $-90°$ and $90°$ are shown.

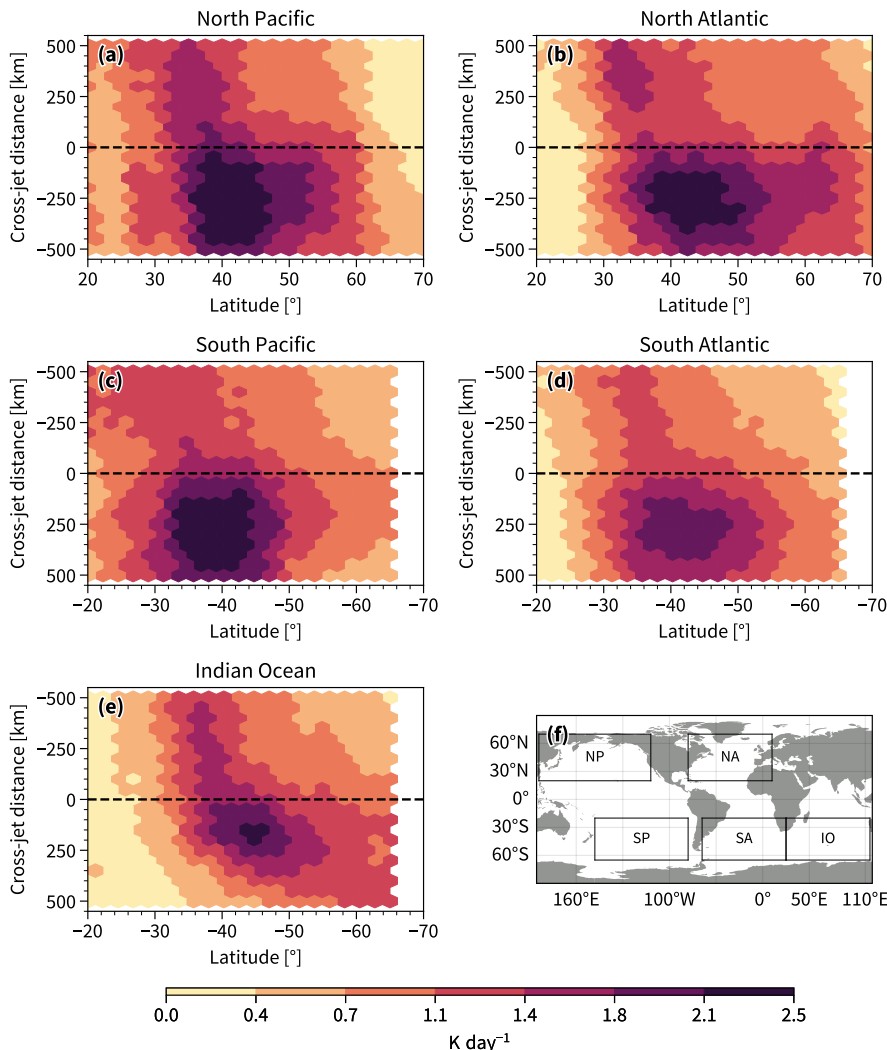

**Figure 8.** Like Fig. 5c for 3h-2D jets, showing latent heating for the five major storm track regions indicated in (f) in their respective winter season. The regions are the same used in Spensberger et al. (2023), namely North Atlantic (80° W–10° E, 20° N–70° N), North Pacific (120° E–120° W, 20° N–70° N), South Atlantic (65° W–25° E, 20° S–65° S), South Pacific (180° W–80° W, 20° S–65° S), and Indian Ocean (25E–115E, 20° S–65° S).

## 8    The vertical structure of across-jet latent heating

We have shown that vertically integrated (200-700 hPa) latent heating preferably occurs on the warm flank of the upper-tropospheric 3h-2D jet and argued that this should strengthen baroclinicity in the jet. Figure 10 shows the vertical distribution of the latent heating composited according to the jet direction. This shows that updraft and latent heating occurs on the warm

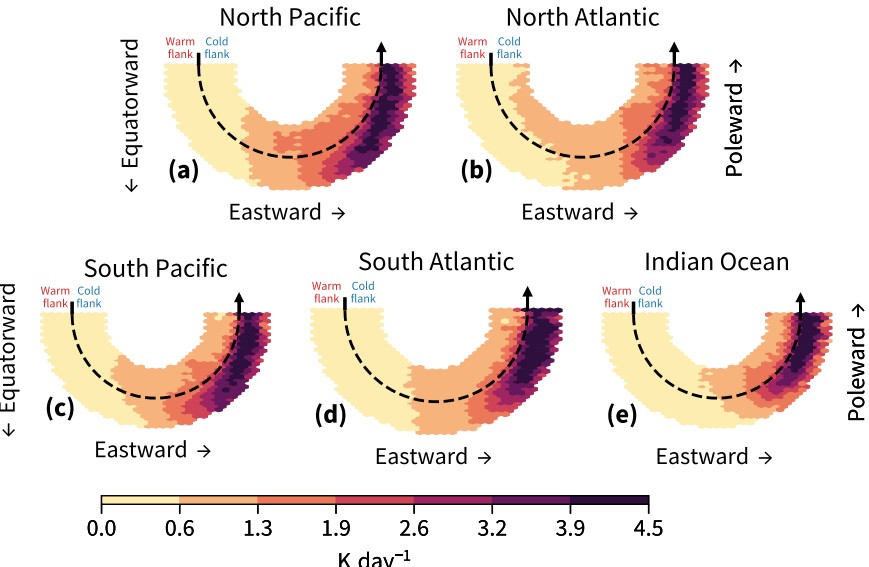

**Figure 9.** Like Fig. 7c for 3h-2D jets, showing latent heating for the five major storm track regions (Fig. 8f) in their respective winter season. The axis direction "poleward" is northward in the NH and southward in the Southern Hemisphere.

side of the jet (left in Fig. 10a,e) from 700 hPa and upwards, leading to an enhanced isentropic slope across the jet (consistent with Papritz and Spengler, 2015; Woollings et al., 2016; Marcheggiani and Spengler, 2023).

There is also a clear signature of diabatic heating on the cold side of the jet at lower levels (below 700 hPa; Fig. 10a,c,e). This is likely caused by surface sensible heat fluxes and subsequent turbulent mixing in the boundary layer. The effect of surface fluxes on baroclinicity is discussed in other works (e.g. Swanson and Pierrehumbert, 1997; Papritz and Spengler, 2015; Marcheggiani and Spengler, 2023).

The differences in equivalent potential temperature (following Bolton, 1980) to the potential temperature on the warm side of the poleward jets is due to water vapour (see right panels of Fig. 10). Consistent with Pauluis et al. (2010), ascent occurs where the difference in lower level equivalent potential temperature to the upper level potential temperature is small, and the the air therefore being close to convectively unstable.

## 9 Concluding remarks

We analysed the across-jet distribution of latent heating, wind speed, humidity, and precipitation for two different types of jet definitions, one based on sector zonal mean zonal wind and one considering the two-dimensional structure of the wind field (Spensberger et al., 2017). We applied these definitions to both low-pass filtered and instantaneous data.

We find that latent heating occurs preferentially on the warm flank of the two-dimensional instantaneous jet (Fig. 7), consistent with Papritz and Spengler (2015) and Woollings et al. (2016), who argued that latent heating mainly acts to increase

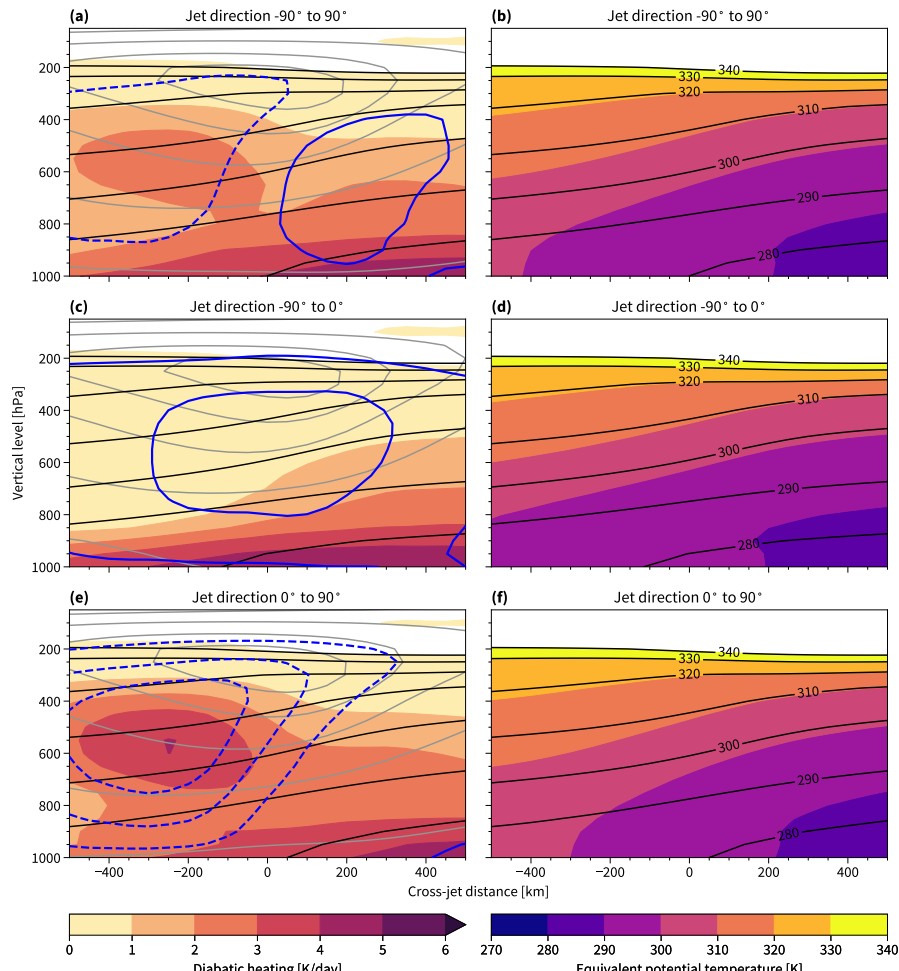

**Figure 10.** Diabatic heating (except radiation; left column; shading), wind in jet direction (grey contours), isentropes (black contours), and vertical wind (blue contours) averaged over composites of the 2D-3h cross sections conditioned on the jet direction. Also shown is the equivalent potential temperature (right column; shading). See Fig. 7 and the text for further explanation of the jet direction. The dashed blue contours show updraft and solid blue contours show downdraft. The contour intervals for vertical wind are -0.1, -0.06, -0.02, 0.02, 0.06, 0.1 $\mathrm{Pa\,s^{-1}}$ and for the wind in the jet direction 5, 15, 25, 35, 45, 55 $\mathrm{m\,s^{-1}}$. Isentropes are identical for the left and right column.

baroclinicity. Conversely, for spatio–temporally averaged jets, the latent heating largely shifts to the cold side of the jet (Fig. 3). This dichotomy is due to latent heating primarily occurring on the warm flank of poleward oriented instantaneous jets, whereas latent heating appears distributed across the jet in a spatial and time mean view. Given that Xia and Chang (2014) base their arguments on spatio–temporally averaged data, the conflicting results with Woollings et al. (2016) are most likely attributable to the sensitivity of the latent heating jet relationship to spatio–temporal averaging.

Consistent with the Clausius–Clapeyron relation, we find that specific humidity is higher on the warm flank of the jet, irrespective of jet orientation and jet definition (Figs. 3,5,7). Vertical wind, on the other hand, is sensitive to the spatio–temporal averaging, which is likely related to upward motion being associated with poleward oriented instantaneous jets and downward motion with the equatorward oriented jets (Fig. 7).

  As latent heating is contingent on moist air being lifted, the across-jet sensitivity of latent heating to spatio–temporal av-
275 eraging thus stems from the asymmetric response in vertical wind to jet direction. This is also reflected in the large-scale precipitation, which exhibits the same sensitivity to spatio–temporal averaging as latent heating and vertical wind (Fig. 6). Hence, variables that are independent of jet direction (e.g., specific humidity) are more robust to averaging techniques than variables that depend on the jet direction (e.g., vertical wind).

  Given that the jet-relative position of latent heating is sensitive to both spatio–temporal averaging and the jet definition, the
280 physical interpretation of the effect of latent heating is similarly sensitive. While latent heating primarily enhances baroclinicity in a two-dimensional instantaneous jet framework, a zonalisation of the jet due to its definition or spatio–temporal averaging yields a reduction in baroclinicity due to latent heating. However, given that most of the latent heating (precipitation) in the mid-latitudes is driven by synoptic features (Konstali et al., 2024), any definition or spatio–temporal averaging of the jet that masks the synoptic nature of the latent heating–jet relation yields a potentially physically misleading interpretation.

*Code and data availability.* The Python library *Dynlib* (Spensberger, 2024) is freely available at https://doi.org/10.5281/zenodo.10471187, and ERA5 data (Hersbach et al., 2020) are freely available at https://www.ecmwf.int/en/forecasts/dataset/ecmwf-reanalysis-v5. The instantaneous two-dimensional jet axes are available at https://doi.org/10.11582/2023.00120, and for the same jets; the cross sections featuring a standard set of variables https://doi.org/10.11582/2023.00121 (Spensberger et al., 2023).

**Appendix A: On the variability of the across-jet latent heating**

As mentioned in Sect. 4, the jet conditional climatologies are coarse grained by binning the data and averaging over the members in each bin. Although the pattern clearly indicates more latent heating on the warm flank of the 3h-2D jets, the spread within each of the bins comprising Figs. 8, 9 is quite large (not shown). Figure A1 summarises the results for 3h-2D jet cross sections by showing the difference in latent heating averaged over the cold side and subtracted from latent heating averaged over the warm side (ΔLH) when total latent heating is distinctively different from zero. There is an overweight of positive
ΔLH occurrences (red), meaning that latent heating mostly strengthen the cross-jet temperature contrast. The distribution of ΔLH is negatively skewed and has similar extreme values on the positive and negative sides.

  We can only trust Figs. 3-6 to be representative of the typical jet-latent heating relationship where the jet sample size is sufficiently large. Note that the number of 10d-Z jets poleward of 60° is particularly sparse and therefore might not be representative for jets there (Fig. A2).

**Figure A1.** Latent heating averaged over the warm flank minus latent heating averaged over the cold flank for the cross sections sampled for the 3h-2D jets for the five major storm track regions (see Fig. 8f) in their respective winter season. Red bars indicate a strengthening ($\Delta LH > 0$) and blue bars a weakening of the cross-jet temperature contrast ($\Delta LH < 0$). Only jet cross sections where the mean latent heating computed over both flanks exceeding $1 \text{ K s}^{-1} \text{ km}^{-1}$ are shown, where N is the number of cross sections satisfying this criterion.

## Appendix B: Easterly jets

Figure B1 gives an overview of the 3h-2D jet directions. The African subtropical jet is most frequently detected, followed by a large number of jet directions between $\pm 45°$ over and downstream of the Gulf Stream. Roughly 90 percent of the jets in the NA are westerlies. The remaining 10 percent are easterlies and are on average weaker than the westerly jets (Fig. B2a). Updraft, moisture, latent heating, and large-scale precipitation are all larger on the warm side (outer circle) compared to the cold side (inner circle) for the poleward oriented jets (Fig. B2b-e). The easterly jet directions in Fig. 1b seem to be associated with the bent back front of the easternmost cyclone. As the 3h-2D jet bends towards the east, the right flank must be warmer

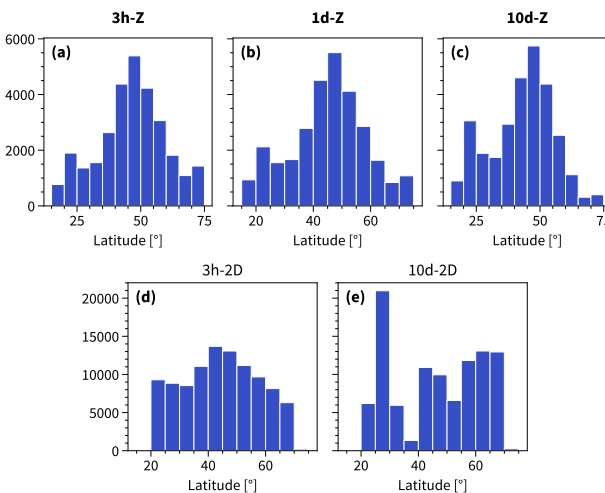

**Figure A2.** Jet sample size as a function of latitude for the five different jet representations (Table 1). Showing the distribution of zonal jet latitudes (top row) and the latitudes of the two-dimensional jet cross-sections (bottom row).

than the left flank by thermal wind, where also the heating occurs. To the north of the bent back front, another westerly jet exists, separating the warmer air over the bent back front from the colder air in the north.

*Author contributions.* HA did the data analysis and took lead on writing the paper. CS provided the cross sections of the instantaneous two-dimensional jets and the infrastructure to compute the 10-day low-pass version. All authors contributed to interpreting the results and to the writing of the paper.

*Competing interests.* At least one of the (co-)authors is a member of the editorial board of Weather and Climate Dynamics.

*Acknowledgements.* We thank the two anonymous reviewers for their helpful comments. We acknowledge ECMWF for the Reanalysis data. The two-dimensional jets were computed using the Python library *Dynlib* (Spensberger, 2024). HA is supported by Aker Scholarship and Laanekassen. PC acknowledges support by UK Natural Environmental Research Council grants NE/V012045/1 and NE/T006250/1. AM, CS, and TS were supported by the Research Council of Norway (NFR) via the BALMCAST project (NFR Project 324081).

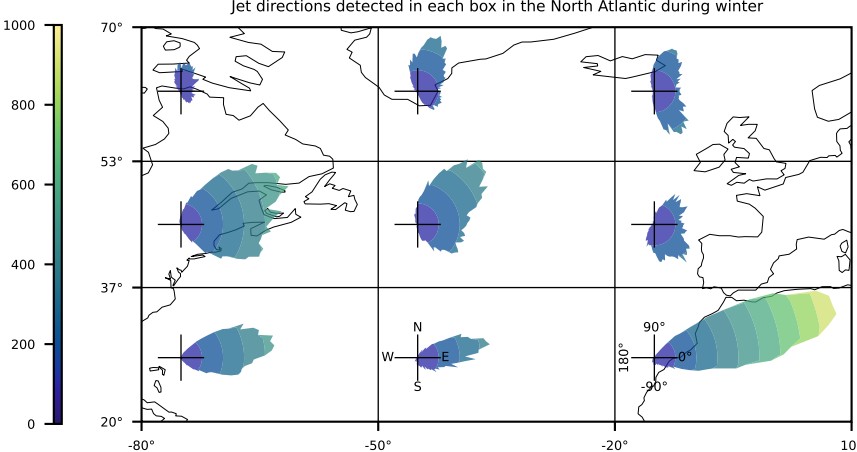

**Figure B1.** Wind rose representation of the 3h-2D jet directions comprising the 100,000 cross sections sampled in the North Atlantic winter. Each wind rose represents the jet directions of the cross-sections in the respective box (black grid). The jet directions are binned in intervals of $1°$ and the colour shading refers the count over each bin. The lower middle and right panels show the cardinal directions and the corresponding jet directions for reference.

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

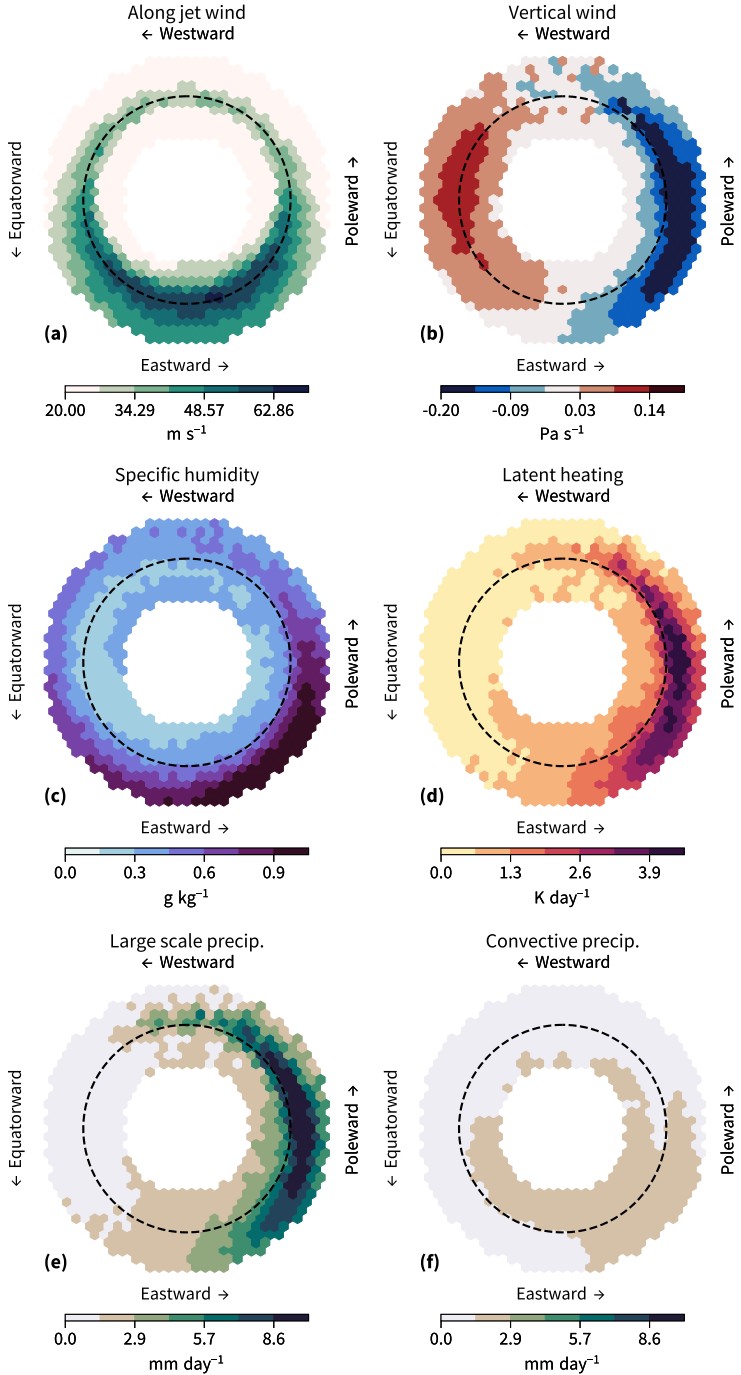

**Figure B2.** Like Fig. 7, but for all jet directions.

Danard, M. B.: On the Contribution of Released Latent Heat to Changes in Available Potential Energy, Journal of Applied Meteorology (1962-1982), 5, 81–84, 1966.

Duchon, C. E.: Lanczos Filtering in One and Two Dimensions, Journal of Applied Meteorology and Climatology, 18, 1016–1022, https://doi.org/10.1175/1520-0450(1979)018<1016:LFIOAT>2.0.CO;2, 1979.

Edmon, H. J., Hoskins, B. J., and McIntyre, M. E.: Eliassen-Palm Cross Sections for the Troposphere, Journal of the Atmospheric Sciences, 37, 2600–2616, https://doi.org/10.1175/1520-0469(1980)037<2600:EPCSFT>2.0.CO;2, 1980.

Harvey, B., Methven, J., Sanchez, C., and Schäfler, A.: Diabatic Generation of Negative Potential Vorticity and Its Impact on the North Atlantic Jet Stream, Quarterly Journal of the Royal Meteorological Society, 146, 1477–1497, https://doi.org/10.1002/qj.3747, 2020.

Hersbach, H., Bell, B., Berrisford, P., Hirahara, S., Horányi, A., Muñoz-Sabater, J., Nicolas, J., Peubey, C., Radu, R., Schepers, D., Simmons, A., Soci, C., Abdalla, S., Abellan, X., Balsamo, G., Bechtold, P., Biavati, G., Bidlot, J., Bonavita, M., De Chiara, G., Dahlgren, P., Dee, D., Diamantakis, M., Dragani, R., Flemming, J., Forbes, R., Fuentes, M., Geer, A., Haimberger, L., Healy, S., Hogan, R. J., Hólm, E., Janisková, M., Keeley, S., Laloyaux, P., Lopez, P., Lupu, C., Radnoti, G., de Rosnay, P., Rozum, I., Vamborg, F., Villaume, S., and Thépaut, J.-N.: The ERA5 Global Reanalysis, Quarterly Journal of the Royal Meteorological Society, 146, 1999–2049, https://doi.org/10.1002/qj.3803, 2020.

Hoskins, B. J. and Valdes, P. J.: On the Existence of Storm Tracks, Journal of Atmospheric Sciences, 47, 1854–1864, 1990.

Hoskins, B. J., James, I. N., and White, G. H.: The Shape, Propagation and Mean-Flow Interaction of Large-Scale Weather Systems, Journal of the Atmospheric Sciences, 40, 1595–1612, https://doi.org/10.1175/1520-0469(1983)040<1595:TSPAMF>2.0.CO;2, 1983.

Konstali, K., Spensberger, C., Spengler, T., and Sorteberg, A.: Global Attribution of Precipitation to Weather Features, Journal of Climate, 37, 1181–1196, https://doi.org/10.1175/JCLI-D-23-0293.1, 2024.

Lachmy, O. and Kaspi, Y.: The Role of Diabatic Heating in Ferrel Cell Dynamics, Geophysical Research Letters, 47, e2020GL090 619, https://doi.org/10.1029/2020GL090619, 2020.

Lee, S. and Kim, H.-k.: The Dynamical Relationship between Subtropical and Eddy-Driven Jets, Journal of the Atmospheric Sciences, 60, 1490–1503, https://doi.org/10.1175/1520-0469(2003)060<1490:TDRBSA>2.0.CO;2, 2003.

Lorenz, D. J.: The Role of Barotropic versus Baroclinic Feedbacks on the Eddy Response to Annular Mode Zonal Wind Anomalies, Journal of the Atmospheric Sciences, 79, 2529–2547, https://doi.org/10.1175/JAS-D-22-0061.1, 2022.

Lorenz, D. J. and Hartmann, D. L.: Eddy–Zonal Flow Feedback in the Southern Hemisphere, Journal of the Atmospheric Sciences, 58, 3312–3327, https://doi.org/10.1175/1520-0469(2001)058<3312:EZFFIT>2.0.CO;2, 2001.

Lutsko, N. J. and Hell, M. C.: Moisture and the Persistence of Annular Modes, Journal of the Atmospheric Sciences, 78, 3951–3964, https://doi.org/10.1175/JAS-D-21-0055.1, 2021.

Madonna, E., Wernli, H., Joos, H., and Martius, O.: Warm Conveyor Belts in the ERA-Interim Dataset (1979–2010). Part I: Climatology and Potential Vorticity Evolution, Journal of Climate, 27, 3–26, https://doi.org/10.1175/JCLI-D-12-00720.1, 2014.

Marcheggiani, A. and Spengler, T.: Diabatic Effects on the Evolution of Storm Tracks, Weather and Climate Dynamics, 4, 927–942, https://doi.org/10.5194/wcd-4-927-2023, 2023.

Martius, O., Schwierz, C., and Davies, H. C.: Tropopause-Level Waveguides, Journal of the Atmospheric Sciences, 67, 866–879, https://doi.org/10.1175/2009JAS2995.1, 2010.

Novak, L. and Tailleux, R.: On the Local View of Atmospheric Available Potential Energy, Journal of the Atmospheric Sciences, 75, 1891–1907, https://doi.org/10.1175/JAS-D-17-0330.1, 2018.

Orlanski, I.: Bifurcation in Eddy Life Cycles: Implications for Storm Track Variability, Journal of the Atmospheric Sciences, 60, 993–1023, https://doi.org/10.1175/1520-0469(2003)60<993:BIELCI>2.0.CO;2, 2003.

Papritz, L. and Spengler, T.: Analysis of the Slope of Isentropic Surfaces and Its Tendencies over the North Atlantic, Quarterly Journal of the Royal Meteorological Society, 141, 3226–3238, https://doi.org/10.1002/qj.2605, 2015.

Pauluis, O., Czaja, A., and Korty, R.: The Global Atmospheric Circulation in Moist Isentropic Coordinates, Journal of Climate, 23, 3077–3093, https://doi.org/10.1175/2009JCLI2789.1, 2010.

Rivière, G.: Effect of Latitudinal Variations in Low-Level Baroclinicity on Eddy Life Cycles and Upper-Tropospheric Wave-Breaking Pro-

375 cesses, Journal of the Atmospheric Sciences, 66, 1569–1592, https://doi.org/10.1175/2008JAS2919.1, 2009.

Robinson, W. A.: A Baroclinic Mechanism for the Eddy Feedback on the Zonal Index, Journal of the Atmospheric Sciences, 57, 415–422, https://doi.org/10.1175/1520-0469(2000)057<0415:ABMFTE>2.0.CO;2, 2000.

Spensberger, C.: Dynlib: A Library of Diagnostics, Feature Detection Algorithms, Plotting and Convenience Functions for Dynamic Meteorology, Zenodo, https://doi.org/10.5281/zenodo.10471187, 2024.

Spensberger, C., Spengler, T., and Li, C.: Upper-Tropospheric Jet Axis Detection and Application to the Boreal Winter 2013/14, Monthly Weather Review, 145, 2363–2374, https://doi.org/10.1175/MWR-D-16-0467.1, 2017.

Spensberger, C., Li, C., and Spengler, T.: Linking Instantaneous and Climatological Perspectives on Eddy-Driven and Subtropical Jets, Journal of Climate, -1, https://doi.org/10.1175/JCLI-D-23-0080.1, 2023.

Swanson, K. L. and Pierrehumbert, R. T.: Lower-Tropospheric Heat Transport in the Pacific Storm Track, Journal of the Atmospheric

Sciences, 54, 1533–1543, https://doi.org/10.1175/1520-0469(1997)054<1533:LTHTIT>2.0.CO;2, 1997.

Weijenborg, C. and Spengler, T.: Diabatic Heating as a Pathway for Cyclone Clustering Encompassing the Extreme Storm Dagmar, Geophysical Research Letters, 47, e2019GL085 777, https://doi.org/10.1029/2019GL085777, 2020.

Wernli, H. and Gray, S. L.: The Importance of Diabatic Processes for the Dynamics of Synoptic-Scale Extratropical Weather Systems—a Review, Preprint, Dynamical processes in midlatitudes, https://doi.org/10.5194/egusphere-2023-2678, 2023.

Wirth, V., Riemer, M., Chang, E. K. M., and Martius, O.: Rossby Wave Packets on the Midlatitude Waveguide—A Review, Monthly Weather Review, 146, 1965–2001, https://doi.org/10.1175/MWR-D-16-0483.1, 2018.

Woollings, T., Hannachi, A., and Hoskins, B.: Variability of the North Atlantic Eddy-Driven Jet Stream, Quarterly Journal of the Royal Meteorological Society, 136, 856–868, https://doi.org/10.1002/qj.625, 2010.

Woollings, T., Papritz, L., Mbengue, C., and Spengler, T.: Diabatic Heating and Jet Stream Shifts: A Case Study of the 2010 Negative North

Atlantic Oscillation Winter, Geophysical Research Letters, 43, 9994–10,002, https://doi.org/10.1002/2016GL070146, 2016.

Xia, X. and Chang, E. K. M.: Diabatic Damping of Zonal Index Variations, Journal of the Atmospheric Sciences, 71, 3090–3105, https://doi.org/10.1175/JAS-D-13-0292.1, 2014.

Yamada, R. and Pauluis, O.: Wave–Mean-Flow Interactions in Moist Baroclinic Life Cycles, Journal of the Atmospheric Sciences, 74, 2143–2162, https://doi.org/10.1175/JAS-D-16-0329.1, 2017.

Zurita-Gotor, P., Blanco-Fuentes, J., and Gerber, E. P.: The Impact of Baroclinic Eddy Feedback on the Persistence of Jet Variability in the Two-Layer Model, Journal of the Atmospheric Sciences, 71, 410–429, https://doi.org/10.1175/JAS-D-13-0102.1, 2014.