# Peer review of "Spatio-temporal averaging of jets obscures the reinforcement of baroclinicity by latent heating"

_EGUsphere, 2024_

## Author Comment (AC1)

**Spatio–temporal averaging of jets obscure the reinforcement of baroclinicity by latent heating - reply**

Henrik Auestad, Clemens Spensberger, Andrea Marcheggiani,
Paulo Ceppi, Thomas Spengler and Tim Woollings.

May 2024

**General comment to the reviewers**

Thank you for taking the time to provide good and constructive feedback. The line numbering in the reply refer to the lines in the revised manuscript showing the difference from the original manuscript.

**Reviewer 1**

This paper analyzes the spatial distribution of latent heating relative to the jet position when this position is estimated using different methodologies. The main conclusion of the study is that the heating is larger on the warm than on the cold side of instantaneous, non-zonal jets, which suggests that latent heating reinforces rather than depletes baroclinicity. This stormtrack self-maintenance suggests in turn that latent processes should increase rather than decrease zonal index persistence, contrary to what has been argued by previous studies.

I found the paper interesting and the results thought-provoking. I can see how the notion of stormtrack self-maintenance proposed by Hoskins and Valdes many years ago would point to a diabatic enhancement of zonal index persistence by latent heating, which I had not previously realized. At the same time, I am struggling to reconcile this result with the notion that meridional latent transport is important for decreasing the mean baroclinicity. Although I am not fully sold on the paper conclusions, the methodology seems sound (except for the concerns raised below) and the results reasonably convincing, so I support publication. However, I have issues with the terminology and some concerns with the methodology as described below.

Main comments:
1) My main issue is semantic. I think "averaging" is much more appropriate than "filtering" in the context in which the authors are using this word. Indeed, the 2D jet "filtering" used in this paper is reminiscent of an isentropic or quasi-Lagrangian averaging, I think. While the conventional Eulerian mean

circulation is thermally indirect, in isentropic coordinates the "upward" motion (diabatic heating) occurs at low latitudes, consistent with the authors' findings for the 2D jet analysis. In my opinion, Eulerian mean is a more appropriate terminology than sector mean and noting the connection of the 2D jet averages with a quasi-Lagrangian/isentropic formalism would benefit the paper. But beyond this connection, even when Eulerian averages are considered the use of the word "filtering" for time averages seems awkward to me.

Thank you for pointing this out. We used "filtering" as a collective term for the temporal low-pass filtering applied to the wind (in both the 2D and the zonal case) and the longitudinal averages taken over the longitudinal sector in the zonal case. The longitudinal averages span $60°$ ($\sim$ 4700 km at 45°N) and will therefore smooth out synoptic variability, thus the "filtering".

We did not intentionally refer to the 2D jet axis detection and the following climatologies as "filtering". However, as you point out, it is sort of a quasi-Lagrangian averaging and we agree that in that context using "filtering" is awkward.

We agree that using "averaging" instead of "filtering" enhances the readability and have changed that notion except where it refers to the temporal low pass filtering only.

The 3h-2D approach is now briefly discussed in the context of an isentropic analysis on line 248-251.

2) I am struggling to reconcile the synoptic notion that much of the condensation occurs along the warm conveyor belt (which the paper eloquently illustrates) with the climate notion that meridional latent heat transport helps decrease the meridional temperature gradient. I wonder if it might be necessary to consider the full 3D picture to reconcile these two views. The authors focus on where the heating occurs relative to the upper-level jet. But because the warm conveyor belt is 3D and fronts tilt with height, the condensation might in fact occur on the warm side of the jet but on the cold side of the lower-trosposphere temperature gradient. Moreover, as Fig. 1 shows that much of the cold front precipitation tends to occur along the actual front/upper level jet. Because the emphasis of the paper is the impact of the heating on the baroclinicity, I think it would be more appropriate to investigate where the heating occurs relative to the lower-troposphere baroclinicity rather than the upper level jet.

This is a great point, thanks. To understand the effect of latent heating on the lower level baroclinicity we have added a section 8 called "On the vertical structure of across-jet latent heating", lines 238-251, where a new figure (Fig. 10) shows the vertical extent of the diabatic heating in the cross-sections around the 2D-3h jets. Figure 10 confirms that latent heating acts to further tilt the isentropes (increase baroclinicity) throughout the vertical extent of the

heating (700 hPa and upwards), consistent with Papritz and Spengler (2015), Woollings et al. (2016) and Marcheggiani and Spengler (2023).

This analysis suggests that part of the notion that the meridional latent heat transport helps to decrease the meridional temperature gradient perhaps comes from considering the effect of averaged latent heating on an averaged background state.

Other comments:

- The rationale for the procedure described starting in line 80 is not fully clear. Although this is described in detail in a previous paper, a clearer explanation here would help the reader.

  Thanks for pointing this out. Line 84-87 hopefully now clarifies the rational for the 2D jet axis detection procedure.

- It is not clear to me if you allow for the existence of more than one jet when computing a sector mean, or you just look at the overall maximum.

  We only allow for one jet, this has now been emphasised, line 74.

- Relatedly, I am surprised by the fact that you get a Eulerian-mean thermally direct circulation when the jet is at very high latitudes. I wonder if there might be more than one jet in these circumstances so that you may be mixing their signals.

  We agree that there is a high probability of a double jet structure when the jets in Fig. 3 are very poleward or very equatorward. The climatologies in Fig. 3 do not discriminate those events, but states that the poleward shifted jets have latent heating on the warm side while the equatorward shifted ones have it on the cold side, consistent with each other.

- Regarding the "dilution" of this pattern for 10d-Z jets compared to 1d-Z jets (line 130), I suspect the number of cases will be much smaller in the former case (the jet may not remain at such high latitudes for a long time), so these results may not be very robust.

  Good point, thank you. There are indeed fewer 10d-Z jets poleward of 65° compared to the 3h-Z and the 1d-Z jets (not shown). Otherwise, the distribution of jets across latitudes are quite similar for the three time filters used. We have made a note of this in line 139-141.

- It is clear from the 10d-2D results that any relation to the actual jet has been lost in this analysis, and all you can see is noise added to a background latitudinal distribution. I would remove this analysis.

  Reviewer 2 had a similar concern, and we repeat the answer also given

to reviewer 2. The 10d-2D analysis is relevant, because: We included the 10d-2D analysis (1) for completeness, (2) because it is indeed a surprising result and (3) to show that the locations of the the 10d-2D jet axes are not representative of an instantaneous wind structure resembling a jet. We think that this, again, is the reason for the lack of sensitivity in the analysed variables to the "presence" of the a 10d-2D jet axis. We have tried to make (3) clear in lines 180-181,194-196.

- Line 181. Figs. 6c and 6d look quite different to me. They only agree in the weak latitude-dependent background.

  Thanks. We were trying to say that the two patterns in convective precipitation (Figs. 6c,d) are more similar to each other than the patterns in large-scale precipitation (Figs. 6a,b). Although the patterns in convective precipitation are largely different in magnitude, they are similar in shape, which suggests that the qualitative relationship between 2D jet and convective precipitation changes little with the 10-day low-pass filter used. Lines 202-205 now clarifies this.

- I believe there is a mistake and the label "Poleward" should be placed on the right of the panels (as in Fig. 2c). This is also confusing, it may be helpful to add an "Equatorward" label on the left axis.

  We think that this is a great suggestion to improve the readability of the jet direction figures. The two labels were initially just thought to replace the the x- and the y-axis labels, but matching those with the location of the jet directions is a good idea. This has been updated.

Minor
61. The second clause is not a full sentence. I suggest using a colon sign instead of a period to separate the first two clauses.

Thank you, the correction has been made. Line 65.

Eq. 1. I don't think n should be hatted in the denominator.

Thank you, sorted.

80. I believe a U is missing in front of the second n-derivative in the equation.

Thank you, sorted. Line 86.

The transition from line 89 to 90 is too abrupt. As written, an example is construed as general. You should say something like "To illustrate the differences between the jet definitions, Fig. 1 shows..."

Thank you, this has be corrected. Line 98-99.

209. Sect. A should be Appendix A.

Thanks. We have corrected this.

**Reviewer 2**
Summary
This manuscript presents an analysis of the observed relation between the upper-tropospheric jet location and midlatitude mid-tropospheric latent heating. The motivation is to examine whether latent heating occurs mostly on the warm/cold side of the jet, thus increasing/decreasing the baroclinicity around the jet peak. Specifically, the authors aim to reconcile contrasting results from previous studies, some of which found that latent heating tends to reduce the baroclinic shear of the zonal-mean or time-mean jet, while others found that overall latent heating tends to reinforce the baroclinicity at midlatitudes. Using spatio-temporal filtering, the authors show that while latent heating occurs mostly poleward of the zonal-mean time-mean jet (Fig.3l), it occurs mostly on the right (i.e., warm) side of the instantaneous local jet (Fig.5g). They further stratify the data according to the instantaneous local jet direction and show that latent heating occurs mostly to the east of poleward-directed jets (Fig.7c). These results are shown for the winter North Atlantic jet, with some result shown also for other longitudinal sectors in both hemispheres, showing qualitative similarity to the North Atlantic jet results (Figs.8,9).

I think these results would be interesting for the Weather and Climate Dynamics readers. The manuscript adds another piece to the puzzle of the interaction between the circulation and moisture (specifically latent heat release) in midlatitudes. I have some comments that I think should be addressed before the manuscript is accepted for publication.

Major comments

- The description of the analysis method in section 4 could be improved. Currently it is somewhat confusing. These are the points that weren't clear to me:

  - Lines 98-101: This description is not clear. I understood it only after I continued to read. Perhaps it would help to explain here what the horizontal and vertical axes in figure 2a represent. It took me a while to understand that "the lowest jet latitude states are on the left and highest jet latitude states on the right" refers to the horizontal axis in figure 2a.

    Great, thanks for making us aware of this. We have added a description of the axes in Fig. 2 in line 109-111. Hopefully this is

clearer now.

– In the description of figure 2b the authors write that (lines 105-108) "we rotate the two-dimensional jets so that they appear as pure westerlies. With this rotation, the cold flanks of the jets are on the positive- and the warm flanks on the negative side of the vertical axis". In the description of figure 2c they write that (lines 110-112) "...we make use of the jet cross sections but this time organize them by the direction of the jet rather than their latitude. Their original orientation in the latitude-longitude space is preserved and the cross sections are thus organized in semicircles according to their orientation". I couldn't understand why in figure 2c (and later in figure 7) the cross sections are organized in semicircles and not in circles. How are westward winds represented? This seems like a critical point for the interpretation of figure 7. If there is latent heating to the right of a westward jet, then this means latent heating on the poleward side of the cyclone.

Good point. A version of Fig. 7 have been added to Appendix B, where the easterly jets are included together with a discussion of the easterly winds. 90 percent of the winds are westerly, now mentioned in lines 223-224. We think a circular representation is less intuitive to understand and might be confused with a cyclone, while there is no guarantee that there is a cyclone in the close vicinity of the jet. Since also there are comparatively few easterly jets, we do not think it is too important for the overall climatology, and therefore justified to focus on the westerly jets in the main body of the paper. Having first gotten an understanding for Fig. 7, it is easier to grasp the full circle representation, and we therefore think that it works best to add this to Appendix B.

– I suggest to replace the wording of the "warm/cold" side of the jet throughout the manuscript to the "right/left" side. There are no figures showing the temperature anomalies in the manuscript. Though we could expect the right side to be warmer than the left side, this is not strictly speaking what the right/left sides are. Specifically, when interpreting the results for the zonal-mean jet (e.g., lines 129-130), it could be that the equatorward (right) side of the jet is not necessarily the warmer side, when the 3h local variables are considered.

Thanks for pointing this out. We have added contours showing the 3h temperature to Fig. 3 and Fig. 5. They confirm that the right side is also the warm side and the left side is the cold side. The

"warm/cold" terminology is consistent in both hemispheres and for all jet directions. We have chosen to stick with the more general terminology and not changed from using "warm/cold" to "right/left" and hope that this is OK! The temperature is discussed in lines 138-140,178,180.

- The interpretation of the results:

  - The analysis of the zonal-mean jet (figures 3,4) shows a diagonal pattern, which I assume represents a constant latitude for the analysis point (for example a point at latitude 40 would be poleward of the jet when the jet is at latitude 20 and equatorward of the jet when the jet is at latitude 50). It would aid the interpretation to add to these figures a diagonal line showing the slope of a constant-latitude line. In this context, I am not sure why the authors claim that "...a banded latitudinal structure ... indicates that is has a strong geographical preference" (lines 148-151, and similarly in line 183). If there is a geographical preference for a specific variable, shouldn't it appear as a diagonal structure in figure 3?

    Correct, constant latitude will be along diagonal lines in Figs. 3 and 4, and we have added such lines to the figures.

    Indeed, it is not clear from Fig. 4 that there is a banded latitudinal structure in the convective precipitation. This is due to the choice of contour intervals, which we have updated. Thanks for pointing this out! Now, it is hopefully more clear that there is a structure of latitudinal constant convective precipitation that intersects the jet axis (black dashed) at 40°N.

  - The comparison with the results of Xia and Chang (2014) in lines 152-156: It should be noted that the data analyzed here is for winter, while Xia and Chang (2014) analyzed summer data. Note that the relation between the latitude of maximum diabatic heating and the jet latitude depends on the season (Lachmy and Kaspi 2020).

    Thank you for pointing out this important detail that we have forgotten to mention. Updated in line 40-41 and line 171. Lachmy and Kaspi (2020) analyse the monthly mean lower tropospheric zonal winds, while we analyse the upper-tropospheric winds. Having shown how sensitive the latent heating jet relationship is to time averaging we wonder whether directly comparing the daily results to Lachmy and Kaspi (2020) is like comparing apples and pears.

  - The structure of the 10d-2D results: I couldn't really understand what this analysis represents and why the patterns in the right column of figure 5 seem to depend only on the jet latitude. Looking at the orange dashed line in figure 1b, I wouldn't a-priori guess that

the analysis variables would depend on the latitude of this line rather than the latitude of the analysis point. If the 10d-2D results are to be included in the paper, they should be explained. Otherwise, consider removing them.

Thanks for pointing this out. We included the 10d-2D analysis (1) for completeness, (2) because it is indeed a surprising result and (3) to show that the locations of the the 10d-2D jet axes are not representative of an instantaneous wind structure resembling a jet. We think that this, again, is the reason for the lack of sensitivity in the analysed variables to the "presence" of the a 10d-2D jet axis. We have tried to make (3) clear in lines 180-181,194-196.

– The interpretation of the 3h-2D results: The authors claim that these results "indicate an enhancement of baroclinicity at all latitudes in the extratropics" (lines 186-188), and contrast these results with previous studies which "highlighted detrimental effects of latent heating on the jet". I think this interpretation adds unnecessary confusion. The instantaneous local (3h-2D) "jet" has a different meaning compared to the zonal-mean time-mean (10d-Z) jet. In the context of wave-mean flow theory, the 3h-2D jet represents mostly the eddies. The results of the current manuscript show that latent heating contributes to the baroclinicity within individual eddies (in other words, to the eddy available potential energy), but not the zonal-mean time-mean baroclinicity (the mean available potential energy). Specifically, it seems that the maximum latent heating appears in the poleward-moving warm conveyor belt inside cyclones. Perhaps the warming occurs on the poleward side of the cyclone (if figure 7 would be a full circle rather than a semi-circle then this would be visible). Therefore, I think that referring to both the 3h-2d and the 10-Z wind maxima by the same word – "jet" – is confusing. These are essentially different components of the flow. The same comment applies to lines 196-199, 230-235 and other places in the manuscript.

This is a really important point to clarify, thanks for bringing attention to this. The 10d-Z analysis indeed shows latent heating relative to a time- and zonal mean background state, while the 3h-2D analysis shows latent heating relative to the 3-hourly winds. Given that the 3h-2D analysis reflect the "instantaneous" relationship between jet and latent heating, it comprise both the eddies and the mean components of e.g. a Reynolds decomposition; if you average the instantaneous state, you get the mean state remaining. As spatio–temporal averaging aggregates instantaneous occurrences, we think that the 3h-2D analysis is relevant for the zonal/time mean baroclinicity. This view is consistent with Papritz and Spengler (2015), where the diabatic generation of baroclinicity is calculated from instantaneous data and then averaged over. We have clarified this in line 212-216.

Parts of the motivation for writing this paper was to reflect on the fact that jets have been defined in countless ways in the literature and to investigate what implications a jet definition have for e.g. the ambient variables and in this case latent heating in particular. In that respect, we think that we have chosen 5 representative ways of defining a jet, that span a range of time and spatial scales. We have tried to consistently distinguish between the different time and spatial scales of the different jets by e.g. consistent use of their labels. Thus, we feel that not using the word jet for the 3h-2D representations would compromise the message of the paper, although the association of "jet" with "mean flow" is perhaps challenged. We have emphasized that 'jet' has a range of meanings in the literature in line 49-50.

- Citations: There are a few relevant citations, which are not discussed in this manuscript:

  - Madonna et al. (2014) – It would add some insight to relate the results in the current manuscript to the view of the warm conveyor belt. This subject is mentioned in lines 194-195, but without this reference, which I think is the most relevant. I would guess that most of the latent heating found in the 3h-2D analysis comes from warm conveyor belts.

    Thanks for pointing to this. We referred to Harvey et al. (2020) as they are discussing WCBs in the context of jets. Of course also Madonna et al. (2014) should be mentioned. Added in line 223.

  - – They analyze the observed climatological zonal-mean structure of the midlatitude diabatic heating and how it is related to the zonal-mean eddy-driven jet. I think this is relevant for the interpretation of the 10d-Z results.

    Thanks! We agree that Lachmy and Kaspi (2020) is relevant for the 10d-Z results. We now discuss this in line 173-175.

  - Pauluis et al. (2010) – This observational analysis looks at diabatic heating (upward motion in potential temperature coordinates) when it is averaged over moist isentropic surfaces, which capture the diabatic heating within eddies (as the 3h-2D analysis here). I think it would add another insight to compare with their results.

    Indeed interesting to compare this to Pauluis et al. (2010). We now do so in line 248-251.

– Yamada and Pauluis (2017) – While this is a numerical study and not observational analysis, I think it would be useful to compare with their results, which show the role of latent heating in the development of baroclinicity during an eddy life-cycle.

Thanks. We now discuss Yamada and Pauluis (2017) in line 33-34.

Minor comments
Line 45: "explore to the extent which" $->$ "explore the extent to which".

Thank you, this has now been corrected. Line 48.

I would suggest to add a plot for the wind speed in figure 7, to make it comparable to the previous figures.

We added wind speed to the new plot also showing the easterlies. Ideally, we would keep the 3x2 structure of Fig. 7, while including panel (d) to make the figure easier to understand.

Other:
Edward Groot kindly pointed out that the regions in Fig. 8 were wrongly described in the figure text. This is now corrected.

**References**

Harvey, B., J. Methven, C. Sanchez, and A. Schäfler (2020). "Diabatic Generation of Negative Potential Vorticity and Its Impact on the North Atlantic Jet Stream". In: *Quarterly Journal of the Royal Meteorological Society* 146.728, pp. 1477–1497. DOI: 10.1002/qj.3747.

Lachmy, O. and Y. Kaspi (2020). "The Role of Diabatic Heating in Ferrel Cell Dynamics". In: *Geophysical Research Letters* 47.23, e2020GL090619. DOI: 10.1029/2020GL090619.

Madonna, E., H. Wernli, H. Joos, and O. Martius (2014). "Warm Conveyor Belts in the ERA-Interim Dataset (1979–2010). Part I: Climatology and Potential Vorticity Evolution". In: *Journal of Climate* 27.1, pp. 3–26. DOI: 10.1175/JCLI-D-12-00720.1.

Marcheggiani, A. and T. Spengler (2023). "Diabatic Effects on the Evolution of Storm Tracks". In: *EGUsphere*, pp. 1–23. DOI: 10.5194/egusphere-2023-1537.

Papritz, L. and T. Spengler (2015). "Analysis of the Slope of Isentropic Surfaces and Its Tendencies over the North Atlantic". In: *Quarterly Journal of the Royal Meteorological Society* 141.693, pp. 3226–3238. DOI: 10.1002/qj.2605.

Pauluis, O., A. Czaja, and R. Korty (2010). "The Global Atmospheric Circulation in Moist Isentropic Coordinates". In: *Journal of Climate* 23.11, pp. 3077–3093. DOI: 10.1175/2009JCLI2789.1.

Woollings, T., L. Papritz, C. Mbengue, and T. Spengler (2016). "Diabatic Heating and Jet Stream Shifts: A Case Study of the 2010 Negative North Atlantic Oscillation Winter". In: *Geophysical Research Letters* 43.18, pp. 9994–10, 002. DOI: 10.1002/2016GL070146.

Xia, X. and E. K. M. Chang (2014). "Diabatic Damping of Zonal Index Variations". In: *Journal of the Atmospheric Sciences* 71.8, pp. 3090–3105. DOI: 10.1175/JAS-D-13-0292.1.

Yamada, R. and O. Pauluis (2017). "Wave–Mean-Flow Interactions in Moist Baroclinic Life Cycles". In: *Journal of the Atmospheric Sciences* 74.7, pp. 2143–2162. DOI: 10.1175/JAS-D-16-0329.1.

---

## Author Response (AR2)

**Spatio–temporal averaging of jets obscure the reinforcement of baroclinicity by latent heating - reply #2**

Henrik Auestad, Clemens Spensberger, Andrea Marcheggiani,
Paulo Ceppi, Thomas Spengler and Tim Woollings.

July 2024

**General comment to the reviewers**

We would like to thank both reviewers for a comprehensive assessment of the manuscript in both rounds of review. We think that your comments have significantly improved the manuscript and provoked interesting ideas and discussions. Please find an answer to all of your comments below. For most comments, we have made changes to the manuscript. In the cases we did not change the manuscript, we have explained why and hope that you find the explanations satisfactory.

The line numbering in our reply refers to the version with tracked changes.

**Reviewer 1**

Summary

The manuscript has been revised based on comments from two reviewers including myself. I am satisfied with most of the revision. I think the results are very interesting and add insight to the role of latent heating in the midlatitude circulation.

Besides a few minor comments, I still have two major suggestions. One suggestion is for an additional figure that could aid the interpretation of the results (see major comment 1). The second suggestion (major comment 2) is regarding the physical interpretation of the difference the authors find between the effect of latent heating on baroclinicity for the averaged and instantaneous flows. I don't disagree with the interpretation given in the manuscript, but I think that using other well-known concepts from the literature could add to the contribution of the paper to the theoretical understanding of the role of latent heating in the midlatitude circulation.

Major comments

1) As explained in lines 124-127, the results presented in this paper are calculated by binning the data and averaging over all members in each bin. It could be that the sample size varies greatly between bins. I think that adding a figure showing the sample size for each bin (for the 3h-Z, 1d-Z, 10d-Z, 3h-2D, 10d-2D and for the 3h-2D sorted by jet direction) would help to interpret the results better. It is important to see which cases are common and which are rare. For example, in lines 140-141 the authors mention that the patterns found at high latitudes in the 10d-Z analysis might not be robust due to the small sample size.

Thank you for pointing this out. We have added the requested figure in Appendix A and changed line 143.

2) As implied in the title and in many places in the manuscript (e.g. lines 209-212, 256-262 and 272-278), the authors argue that latent heating reinforces baroclinicity and that averaging over longitude and time obscures this picture. I agree that this statement is consistent with the results, but I think it gives a misleading impression for the role of latent heating. Looking for example at the lower panel of figure 1b, it is clear that latent heating occurs inside the warm conveyor belt of a frontal cyclone. The "jet" in this case is the upper level northward wind inside the cyclone, which is maximal slightly westward of the latent heating that occurs in the mid-troposphere. It is true that strictly speaking this reinforces the baroclinicity inside the cyclone. However, the term "baroclinicity" usually refers to the background (averaged) flow, which may become baroclinically unstable and allow for further growth of eddies (mostly cyclones). Here I think this local baroclinicity represents the available potential energy of the cyclone and not the background flow. One can think of it in the context of the Lorenz energy cycle: mean available potential energy (MAPE) is converted to eddy available potential energy (EAPE) which is then converted to eddy kinetic energy (EKE). I would argue that the results of this paper demonstrate that latent heating reduces MAPE and increases EAPE. Therefore it plays a role in the energy transfer from the mean flow to the eddies. This interpretation is based on the following results:

(a) Latent heating is located eastward of the upper level northward instantaneous jet (Figures 5g, 7c, 9, and the example in Figure 1).

(b) The regions of strong latent heating are characterized by upward wind, large-scale precipitation, positive temperature anomaly and high specific humidity (Figures 5-7).

(c) Latent heating is concentrated on the poleward side of the zonally- and temporally-averaged jet (Figure 3j,k,l).

Results a and b fit well with the characteristics of frontal cyclones and warm conveyor belts. Inside the cyclone there is a positive correlation between temperature and upward wind ($w'T'>0$) and temperature and northward wind ($v'T>0$), as well as specific humidity and upward or northward wind ($w'q'>0$, $v'q'>0$). The positive correlation between temperature and northward wind ($v'T'>0$) is

necessary for the conversion of MAPE to EAPE and the positive correlation between temperature upward wind ($w'T'>0$) is necessary for the conversion of EAPE to EKE (see Lorenz, 1955; and a recent example in Okajima et al. 2022). Latent heating then contributes to this temperature anomaly and to the conversion of MAPE to EAPE. Note that in this paper the "jet" is measured at a higher level than the latent heating so that if the flow inside the cyclone is tilted westward with height (as in a characteristic baroclinically unstable eddy), then the upper level northward wind would be slightly westward of the mid-tropospheric northward wind and the positive temperature (and latent heating) anomaly that is positively correlated with it. Also note that the opposite patterns of upward/downward motion between the instantaneous and averaged jets (lines 182-184) are consistent with the differences between the Ferrel circulation and the Lagrangian-mean circulation (as approximated by the Transformed Eulerian Mean). The fact that latent heating is concentrated on the poleward side of the zonally averaged jet (figure 3) demonstrates that it acts to decrease MAPE. Therefore, I don't think that saying that latent heating reinforces the baroclinicity delivers the right impression for those thinking in terms of wave-mean flow interaction. My suggestion is to use the well-known concepts of the Lorenz energy cycle to explain the role of latent heating in the midlatitude circulation. The authors can consider whether to accept this suggestion or not.

Thank you for bringing up this important point. Reviewer 2 raised a similar concern in the first round, which was also discussed among the authors before first submitting the paper.

Explaining results (a), (b), and (c) using MAPE and EAPE requires separating eddies from a background (or mean) state and choosing a reference state. Based on our findings, the subjectivity in these actions potentially obscures the link between APE, baroclinicity, and latent heating.

Furthermore, the representation of baroclinicity in the APE framework is highly sensitive to the choice of reference state. While a reference state computed from a channel model scaled for baroclinic instability gives an APE distribution that represents lower tropospheric baroclinicity (Federer et al. 2024), APE relative to a global reference state does not (Novak and Tailleux 2018).

Hence, we find it difficult to reconcile the APE framework in our discussion on the latent heating effect on baroclinicity and have therefore opted to not delve further into this analysis. However, as this discussion is likely of interest to the readers of WCD, we have included a paragraph on lines 239-246 and removed the paragraph on lines 218-220.

Minor comments
1) Line 48: two comments: (a) "explore the extent to which" instead of "explore to the extent which". (b) "... occurs on the cold or warm flank of each jet definition" instead of "... occurs on the poleward or equatorward flank of

each jet definition". For the instantaneous jet its really mostly the westward vs eastward flank.

(a) Thanks and (b) great point, thanks a lot - this has been corrected. Lines 48 and 49.

2) Line 74: "only allow for one jet latitude for each time step" (add "for each").

Thanks - done! Line 75.

3) Line 79: since you removed the hat above n in equation (1), it needs to be removed here as well.

Yes, that was a slip. Done. Line 80.

4) The paragraph in lines 83-91: The parameter K is not defined. Shouldn't it have units? It is written here as if it is a dimensionless parameter, but it should have units of $\sec^{-2}$ if n has units of length. Also there is some confusion with the sign. If you use it to detect a maximum point then K should be negative and the inequality in line 86 should be opposite (the second derivative is lower than some negative threshold). In line 96 it has a positive value, which seems inconsistent.

You are absolutely right, thanks a lot for spotting this! Line 95.

5) Line 91: "de tails" – change to "details" (delete the space).

Done, thanks. Line 90.

6) Lines 99-100: "Fig. 1a shows..." (add "a") and remove "(Fig. 1a)" at the end of the sentence.

Done. Line 99.

7) Lines 118-119: Here it should be "left side" and "right side" instead of "cold flanks" and "warm flanks". It can be explained that the left side is cold and the right side is warm, due to thermal wind balance, and then the terms "cold flanks" and "warm flanks" can be used for the rest of the paper, but since this sentence describes the geometry of the analysis I think the terms "right" and "left" should be used here.

Great suggestion, this has been updated. Lines 118-120.

8) Lines 185-186 and 188-189: "ascent appears to be mainly a function of latitude" and "the specific humidity is... mostly a function of latitude" – it should be "mostly a function of the jet latitude", because if it were a function of latitude, it would have had a diagonal structure, as in figure 3.

Nicely spotted, has been fixed. Lines 192, 195.

9) Caption of figure 5: I think the last sentence should be moved to the methods section.

Thanks, done. Figures 3,5,7 all had similar sentences at the end of their captions, which have all now been moved to the methods. Lines 111-112,120-121,126-129

10) Line 223: "Figure 7 only shows" instead of "only show".

Thanks, done. Line 228.

11) Line 237: "Appendix A" instead of "Sect. A".

Thanks, done. Line 250 (and 143).

12) The caption of figure B1 is not clear. I couldn't understand what it means that the circles correspond to a certain number of cross sections. How is the center and radius of each circle calculated? How is the red contour calculated

Thanks. We have updated Figure B1 and the caption. Hopefully, it is better explained now.

**Reviewer 2**

This is an intriguing piece of work. The authors address the issue of where the stormtrack latent heating occurs relative to the jet, and hence whether this heating reinforces or depletes the extratropical baroclinicity. The analysis helps reconcile two seemingly conflicting views: the climatologist view that the latent heat transport helps decrease the meridional temperature gradient, and the synoptician view that the latent heating actually occurs on the warm sector of extratropical cyclones. The authors convincingly show that both are true. The heating occurs on the poleward side of the Eulerian mean temperature gradient (and hence helps decrease that mean gradient) but on the warm side of the distorted isotherms, consistent with common synoptic experience. One could say that the 2D averages considered in this paper have a bit of a cuasi-Lagrangian flavor.

I feel that the analysis presented in this paper has broader implications than the annular mode persistence problem that seems to be its primary motivation. While also relevant for that problem, the relation between baroclinicity forcing and eddy momentum flux memory is somewhat speculative and still under debate. I suspect the motivation of the authors to emphasize the persistence implications is novelty, as the idea that the stormtracks are self-maintained by their associated latent heating is certainly not new. But even if the notion is not entirely new, I think the analyses in this and previous papers by the same authors provide novel support and physical intuitition for the self-maintenance theory.

The link between self-maintenance (depletion) and persistence as suggested by Xia and Chang (2014) motivates studying the latent heating effect in context of jets that shift meridionally and not only in the climatological storm track. Indeed, the link between baroclinic forcing and eddy momentum flux forcing is debated. We are currently working on quantifying the effect of latent heating on the flow on time-scales beyond the life time of a cyclone, but that has to be for a future publication.

I already provided a first review during the open discussion period in the egusphere. The authors have satisfactorily addressed the most important issues raised in that review. Reading the revised version has raised additional thoughts, most minor. The only comment that may require significanty work is the first one but I leave that at the discretion of the authors as I submit a recommendation of minor revision.

1) I believe most of your diagnostics employ full fields (as opposed to anomalies). Have you actually tried using anomalies instead? That should make the main results cleaner, I think. In particular, the latitudinal-dependent background present in some of the figures might be removed if you subtracted the climatology before compositing. The connection with Xia and Chang (2014) would also be clearer using anomalies because these authors use regressions. Though not

required for acceptance, I urge you to give this some consideration.

Thanks for this suggestion! We have not tried using anomalies, but we agree that the connection to Xia and Chang (2014) would be clearer. Using anomalies, one could e.g. see whether: given the presence of a jet, do we have more or less latent heating than normal, and on which flank. However, if one where to give graphical illustration of the eddy correlation between air temperature and latent heating, we would somehow need to define an anomalous 2D jet axis - which is not straight forward and requires choosing a reference state. Although it would be interesting, redoing the analysis using anomalies would require quite a bit of work and given that our results are already quite clear, we think that this is something that could maybe be left for future work.

2) Fig.3 caption. Please specify which latitudes the diagonal lines correspond to

Thanks, this has been updated in the figure caption.

3) 173. My understanding of the results of Lachmy and collaborators (I'm not one of them) is that *heating* displaces the eddy driven jet equatorward of the eddy heat flux maximum. To shift the eddy driven jet poleward of the heat flux maximum one would need cooling. When the heating/cooling is weak, the surface westerlies and heat flux maximum should be collocated, I think.

This is also our understanding. We have exchanged "poleward/equatorward shifts" with "latitudinal shifts relative to the eddy heat flux" in our reference to this paper to avoid any confusion. Thanks! Lines 178-181.

4) 180. As I said in my previous review, I don't think the 10d-2D results mean much. It is of course your call whether to include them or not but if you want to investigate the sensitivity to the averaging period, I would rather suggest trying some intermediate value like the 1d averaging considered for the sector means. It seems strange that you only consider this averaging in that case. Do you get mixed signals wth the 2D jets perhaps?

We never calculated a 1d-2D version. Time-averaging 40 years of reanalysis data, applying the T84 truncation, detecting the jets, creating a random sample of jet axes, and interpolating all variables to the jet cross-section is quite time-consuming and costly. When choosing how to spend our resources, we choose to compute the "ends of the spectrum" of the 2D jet definitions. It would indeed be interesting to try to see whether a 2D version of daily data is sufficient to capture the 3h-2D signal. We agree that the 10d-2D results do not mean much, but that is also the point. However, we think that the current set of results is sufficient and leave a 1d-2D version for future work. Thanks for the suggestion, though!

5) 190. Could the cold side latent heating occur because the detected jet is subtropical and not associated to cyclones?. Is this heating associated to eastward oriented jet perhaps? I also wonder if the convective precipitation found in this case (line 199) could be associated to tropical/subtropical cyclones rather than extratropical cyclones.

Interesting suggestion. Since we're detecting jets on the 2PVU surface, we doubt that the African easterly jets are well represented, if this is what you are thinking of. Figure 1 in this document shows subtropical 3h-2D jets, identified using the potential temperature threshold of Spensberger (2024). The strongest signal is in the large scale precipitation on the warm flank, but note that the sample size is smaller here.

6) 234. Sect A should be *Appendix* A.

Thanks, done. Line 250.

**References**

Federer, M. et al. (2024). "On the Local Available Potential Energy Perspective of Baroclinic Wave Development". In: *Journal of the Atmospheric Sciences* 81.5, pp. 871–886. DOI: 10.1175/JAS-D-23-0138.1.

Novak, L. and R. Tailleux (2018). "On the Local View of Atmospheric Available Potential Energy". In: *Journal of the Atmospheric Sciences* 75.6, pp. 1891–1907. DOI: 10.1175/JAS-D-17-0330.1.

Okajima, S., H. Nakamura, and Y. Kaspi (2022). "Energetics of Transient Eddies Related to the Midwinter Minimum of the North Pacific Storm-Track Activity". In: *Journal of Climate* 35.4, pp. 1137–1156. DOI: 10.1175/JCLI-D-21-0123.1.

Spensberger, C. (2024). *Dynlib: A Library of Diagnostics, Feature Detection Algorithms, Plotting and Convenience Functions for Dynamic Meteorology.* Zenodo. DOI: 10.5281/zenodo.10471187.

Xia, X. and E. K. M. Chang (2014). "Diabatic Damping of Zonal Index Variations". In: *Journal of the Atmospheric Sciences* 71.8, pp. 3090–3105. DOI: 10.1175/JAS-D-13-0292.1.

[Figure]

Figure 1: Like Fig. B2 in the manuscript but only considering jets with potential temperature in their core on the 2PVU surface equal to or larger than 335 K.